# Gel-Forming Soil Conditioners of Combined Action: Laboratory Tests for Functionality and Stability

**DOI:** 10.3390/polym14214665

**Published:** 2022-11-01

**Authors:** Andrey V. Smagin, Viktor I. Budnikov, Nadezhda B. Sadovnikova, Anatoly V. Kirichenko, Elena A. Belyaeva, Victoria N. Krivtsova

**Affiliations:** 1Soil Science Department, Eurasian Center for Food Security, Lomonosov Moscow State University, GSP-1, Leninskie Gory, 119991 Moscow, Russia; 2Institute of Forest Science, Russian Academy of Sciences (ILAN), 21 Sovetskaya, Uspenskoe, 143030 Moscow, Russia

**Keywords:** synthetic polymer hydrogels, water retention, thermodynamic water potential, surface energy, specific surface area, hydraulic conductivity, soil aggregation, basal respiration, biodegradation, half-life of polymers, mathematical modeling

## Abstract

The research analyzes technological properties and stability of innovative gel-forming polymeric materials for complex soil conditioning. These materials combine improvements in the water retention, dispersity, hydraulic properties, anti-erosion and anti-pathogenic protection of the soil along with a high resistance to negative environmental factors (osmotic stress, compression in the pores, microbial biodegradation). Laboratory analysis was based on an original system of instrumental methods, new mathematical models, and the criteria and gradations of the quality of gels and their compositions with mineral soil substrates. The new materials have a technologically optimal degree of swelling (200–600 kg/kg in pure water and saline solutions with 1–3 g/L TDS), high values of surface energy (>130 kJ/kg), specific surface area (>600 m^2^/g), threshold of gel collapse (>80 mmol/L), half-life (>5 years), and a powerful fungicidal effect (*EC*_50_ biocides doses of 10–60 ppm). Due to these properties, the new gel-forming materials, in small doses of 0.1–0.3% increased the water retention and dispersity of sandy substrates to the level of loams, reduced the saturated hydraulic conductivity 20–140 times, suppressed the evaporation 2–4 times, and formed a windproof soil crust (strength up to 100 kPa). These new methodological developments and recommendations are useful for the complex laboratory testing of hydrogels in small (5–10 g) soil samples.

## 1. Introduction

Gel-forming polymeric materials can be considered as the most promising agents for soil conditioning [1,2,3,4,5], especially for widespread Arenosols [6]. The water absorption of some synthetic hydrogels, for example those based on an acrylic polymer matrix, reaches 500–1000 kg H_2_O per 1 kg of dry polymer and more [7,8,9]. This technological property of acrylic superabsorbents is effectively used in increasing the water-retention of Arenosols for saving water in arid irrigation farming and landscaping [3,4,5,7,8,9,10,11,12,13,14,15,16,17,18,19,20]. Small doses of gel conditioners (0.1–0.6% per mass) are sufficient to improve the water-retaining capacity of sands to the level of fertile loamy soils [13,15,17]. The same and even smaller doses of gel-forming polymeric materials effectively aggregate the sandy particles and protect them from wind erosion [3,4,5,21]. An equally promising direction is the use of gel-forming materials as agents for controlled release systems for agrochemicals and pesticides [1]. Nanostructured synthetic and biopolymer materials [1,2,3,4,22,23] are being actively introduced, as well as smart gels that change their structural organization and release retained biologically active substances under the influence of temperature, humidity, pH and other environmental factors [24,25].

However, despite many promising scientific developments in this area, the actual implementation of gel-forming materials for soil conditioning in modern agriculture and landscaping is constrained by a number of serious issues. These include, first of all, a rather high cost (especially for synthetic superabsorbents) and also the effect of negative soil and environmental factors (biodegradation, osmotic stress from dissolved electrolytes, limiting the rate and degree of swelling in a closed porous space, viscous flow and leaching, contrast of the temperature regime with the interphase transitions, etc.), which drastically reduces the efficiency and profitability of gel-forming soil conditioners [20,26,27,28,29,30]. Most of the modern scientific developments in this area, unfortunately, neglect these problems and focus only on the chemical synthesis of new materials and the laboratory analysis of some of their properties [1,2,4,5,7,8,9,10,11,12,13,14,15,16,17,18,19,20,21,22]. As a rule, only the positive effect of conditioners on the soil is analyzed, and the corresponding negative effects of polymeric conditioners on soil factors are completely ignored [7,8,9,10,11,12,13,14,15,16,17,18,19,20,21,22]. Comprehensive field trials of synthesized materials under real soil and climatic conditions are very rare [1,2,3,4]. An effective system of instrumental methods for assessing the technological properties of gel-forming soil conditioners and their compositions with soil substrates, as well as their dynamics in the soil environment, has not been sufficiently developed and substantiated. All these problems obviously require collective efforts and knowledge not only from chemistss–technologists, but also from soil scientists, ecologists, agronomists, foresters, phytopathologists, mathematical modeling specialists and other scientists with appropriate methods and scientific approaches for a comprehensive assessment of the «soil–gel–plant» system and its dynamics. The result of such efforts should be the synthesis and selection of the most effective complex gel-forming soil conditioners combining the properties of water superabsorbents, improvers of soil dispersity and aggregate structure, erosion resistance, as well as agents for controlled release systems for agrochemicals and pesticides with sufficient resistance to negative environmental factors.

We tried to solve some of these problems in a multi-year research project aimed at developing effective and inexpensive gel-forming soil conditioners with a combined action for agriculture and landscaping [3]. This publication presents a line of innovative soil conditioners with an analysis of their composition and certain mandatory, from our point of view, technological properties using an original system for laboratory study of these properties. The key to this innovative development consisted in the use of a patented technology [3] for the synthesis of combined gel-forming soil conditioners with the filling of the acrylic polymer matrix with natural amphiphilic ingredients (dispersed peat, humates), as well as the introduction of microelements and anti-pathogenic protection agents in the form of silver ions, nanoparticles, and an organic fungicide based on Azoxystrobin. This prototype uses basic technology of copolymerization in solution or free radical-initiated polymerization of acrylic acid salts with acrylamide and a crosslinking agent, mainly used in the production of superabsorbents [5]. Since the introduction of amphiphilic ingredients and electrolytes into the hydrophilic polymer matrix could affect the technological properties both positively and negatively, the purpose of this study was a comparative laboratory analysis of the main functional indicators of the new gel-forming soil conditioners and their hydrophilic prototypes. For this purpose, we proposed original technological criteria and instrumental methods for their assessment in pure hydrogels and in gel compositions with soil substrates. The main research tasks included:thermodynamic assessment and modeling of the water retention of soil substrates under the influence of the gel-forming conditioners;analysis of the saturated and unsaturated hydraulic conductivity of soil substrates with the synthetic gel structures;assessment of the dispersity, structural organization and aggregation strength under the influence of gel-forming conditioners;study of the kinetics of swelling and freezing point dynamics in the synthetic gel structures;assessment of the resistance of the gel-forming polymeric materials to biodegradation;analysis of the fungicidal properties of synthetic gel structures.

The scientific novelty consisted not only in the design of new gel-forming polymeric materials for complex soil conditioning, but also in a comprehensive instrumental analysis of their technological properties and resistance to negative environmental factors. The most important technological result obtained for new gel-forming materials was the successful combination of high water retention, increased dispersity and aggregation of soil particles, reduction of unproductive water losses during evaporation and infiltration, and effective anti-pathogenic protection, along with its own environmental resistance.

## 2. Materials and Methods

### 2.1. Gel-Forming Soil Conditioners and Their Composition

New gel-forming soil conditioners of combined action were synthesized at the Ural Chemical Plant (Russian Federation, Perm) under the Aquapastus trademark and using our patented technology [3]. An obligatory hydrophilic component of the new composite materials is a polymer matrix based on acrylamide (AA) and salts of acrylic acid (Ak) represented by ammonium acrylate and sodium acrylate in different ratios of copolymers. This polymer base provides the foundation for the gel structures in pure aqueous and slightly mineralized solutions, with a high water absorption from 300 to 1000 H_2_O per 1 g of dry material. Filling the polymer matrix with biocatalytic wastes (microbial cells, cell agglomerates and filterperlit), amphiphilic agents (dispersed peat, humates), microelements, biological activity inhibitors, and antipathogenic protection agents (silver ions and nanoparticles, organic fungicides) reduces the cost of synthetic materials with 20–25% and gives them the necessary technological qualities for complex soil conditioning. In the base hydrophilic material, Aquapastus-11 (the A11 gel), an acrylic polymer matrix with a ratio of AA/Ak from 25/75% to 40/60%, is filled by 10–30% (mass) biocatalytic wastes from the production of polyacrylamide. The various modifications include 12% additives of potassium and sodium humates (the A11H gel), along with humates, −0.4% magnesium and zinc, as tracer elements for mineral nutrition (the A11HMZ gel), 0.3–1.3% silver ions in the form of nitrate (the A11Ag gel) or 1% fungicide Azoxystrobin adsorbed on filterperlite (the A11Az gel), as inhibitors of polymer biodegradation and agents for anti-pathogenic protection. In another innovative material, Aquapastus-22 (the A22 or «black» gel), a similar polymer matrix was filled with 10–30% (mass) dispersed peat, as the most accessible and cheap Russian natural biopolymer with the amphiphilic properties necessary for soil aggregation and retention of organic pesticides. This hydrogel could also take a more stable form as a pressure gel structure of the reinforced type due to the fine-dispersed filler. Its two modifications (the A22Ag and the A22Qv gels) contained technological additives to the polymer matrix in the form of 0.3–1.3% of silver ions and nanoparticles, as well as 1% Quadris fungicide (Syngenta-group, Basel, Switzerland) based on Azoxystorobin. This fungicide is one of the few pesticides for intra-soil application that is admissible by the legislation of the Russian Federation. The innovative composite materials were compared with each other, as well as with the known brands of acrylic hydrogels Aquasorb (SNF-group, https://www.snf-group.com; accessed on 26 October 2022) and Zeba (UPL-group, https://www.upl-ltd.com; accessed on 26 October 2022) based on polyacrylamide, acrylic acid and starch.

### 2.2. Mineral Soil Substrates

Since the maximum effect of hydrogels is usually observed in sandy soils, we selected the following three coarse-textured mineral substrates (Table 1) for laboratory testing of the new soil conditioners. Sample № 1 was monomineral fine-grained quartz sand without organic carbon (*C*_org_ = 0%), with a close to neutral reaction of the liquid phase (pH = 6.8) and the absence of electrolytes (electroconductivity EC = 0.2 dS/m); sample № 2 was polymineral loamy-sandy Arenosol from the Karakum Desert (Repetek Nature Reserve, N 38.698789, E 63.227950) with a low content of organic carbon (*C*_org_ = 0.2%), slightly alkaline reaction (pH = 7.6) and a low content of electrolytes in the liquid phase (EC = 1.6 dS/m); and sample № 3 was carbonate loamy-sandy Arenosol from Dubai Horticultural Station (N 25.236850, E 55.325432) and also with a low content of organic carbon (*C*_org_ = 0.3%) and slightly higher alkalinity (pH = 8.1) and salinity of the liquid phase (EC = 5.1 dS/m).

### 2.3. Preparation of Gel-Forming Materials

To exclude the influence of the particle sizes of the hydrogels on their technological parameters, we used the same particle range, from 0.25 to 1 mm for all compared materials, using mechanical grinding and sieving. These samples were used to prepare soil–gel compositions, as well as to study the swelling of the hydrogels themselves in distilled water and saline solutions (potassium chloride) of different concentrations with the possibility of free swelling or its limitation by a given load (pressure). The soil–gel compositions for laboratory analyses, containing doses from 0.1 to 0.3% (mass) of dry gel-forming soil conditioners in the mineral substrates, were prepared by mechanical blending of the sandy substrates and pre-calculated amounts of hydrogels, preliminarily swollen in distilled water (1:100). This method, unlike dry blending, guaranteed a uniform distribution of soil modifiers in small doses in the mineral substrates. A laboratory assessment of the protective properties of the gel compositions with 1% biocide agents incorporated (silver or synthetic fungicide) was performed on a potato-dextrose agar medium in the following gel/agar ratios: 1:1000, 1:200, 1:100, 1:50, and 1:20, which made it possible to obtain biocidal agent concentrations of 10, 50, 100, 200, and 500 ppm.

### 2.4. Methodological Guidelines for Laboratory Testing of Gel-Forming Polymer Cnditioners in Soils

The methodological task of our long-term research consisted in the formation of a universal system of indicators and methods for instrumental assessment of the most important technological properties of hydrogels and soil–gel compositions, as well as their stability in soils. This system included the following parameters that are obligatory, in our opinion, for a fully-fledged laboratory analysis of gel-forming polymeric materials for soil conditioning:

#### 2.4.1. Water-Retention Characteristic

Thermodynamic assessment of water retention is based on the experimental dependence of water content and its thermodynamic potential (water retention curve, WRC). To obtain his, we combine the centrifugation method with modifications from [31] and our new thermo-desorption method [32]. In the centrifugation method, water is removed from the analyzed samples using gravitational and centrifugal physical fields. The work of removing water or the matrix thermodynamic potential is calculated depending on the rotation speed (*n* [rpm]) and geometrical parameters of the centrifuge, according to [31]:undistributed sample: |Ψ| = (0.011·*n*^2^·*R*·cos(θ) + *g*·sin(θ))·*h*,(1)
distributed sample: |Ψ| = 0.0055·*n*^2^·(*R*_2_^2^ − *R*_1_^2^)·cos(θ) + *g*·*h*·sin(θ),(2)
where *R*_1_, *R*_2_, *R* [m] are the distances from the axis of rotation to the surface, bottom and center of mass of the sample with a height of *h* [m]; *g* = 9.81 [m/s^2^] is the acceleration of gravity, and θ [radians] is the rotor angle of inclination.

After centrifugation of liquid water, it is convenient to remove the remaining strongly bound (adsorbed) water through gradual heating with a sequential increase in temperature [32]. The corresponding work or thermodynamic potential of water is related to the drying temperature (*T*) and thermodynamic parameters of the air in the laboratory using the following fundamental equation, as obtained in [32]:|Ψ| = *Q* − β·*T*,(3)
where β = {*Q*/*T_r_* − *R*·ln(*f*)/*M*}, *Q* = 2401 ± 3 kJ/kg is specific heat of evaporation for the temperature range of 0–100 °C; *R* = 8.314 J/(mol·K) is the universal gas constant; *T* [K] is the absolute temperature in the drying oven; and *M* = 0.018 kg/mol is the molar mass of water.

We used centrifuges CLN-16 (Russia), OHAUS FC5515R (Switzerland) and Sigma 2-KHL (Germany), with a speed range from 100 to 12,000 rpm or a corresponding range of absolute values of the thermodynamic potential of water from 0.4 to 2800 J/kg. Differential drying of samples using a moisture analyzer AND MX-50 (Japan) made it possible to estimate the sorption part of WRC up to the absolute values of water potential of 700,000–900,000 J/kg, corresponding to a standard drying temperature of 105 °C and air humidity in the laboratory from 30 to 90%, according to the fundamental Equation (3).

#### 2.4.2. WRC-Calculated Criteria of Soil Structure, Dispersity, Water Retention Energy and Capacity

Approximation of experimental WRC data using the empirical van Genuchten model [33] and the fundamental model of the Deryagin disjoining pressure [34,35] made it possible to calculate the structural curves of pore size distribution, the specific surface area and some energy parameters of interfacial interactions in the tested samples. The formula for calculating the pore volume distribution with the size of their radii (*V*(*r*)) from the van-Genuchten model is as follows [34,36]:(4)$$V(r)=\left(n-1\right)\left(W_{s}-W_{r}\right)\rho_{b}{\left(1+{\left(\alpha \left|P\right|\right)}^{n}\right)}^{-m-1}{\left(\alpha \left|P\right|\right)}^{n}$$
where |*P*| = ρ_ℓ_ |Ψ| is the modulus of equivalent water pressure [Pa], ρ_ℓ,_ ρ*_b_* = ρ*_s_*/(1 + *W*_s_ρ*_s_*/ρ_ℓ_) are water density and soil bulk density [kg/m^3^]; ρ*_s_* = 2650 kg/m^3^ is the density of quartz as the dominant mineral component in the sands; *W_s_* [kg/kg] is the water content in a state of saturation of the soil (total water capacity); *W_r_* is residual moisture content corresponding to a tightly bound water; and α [Pa^−1^], *n*, *m* = 1 − 1/*n* are the empirical van Genuchten constants.

The fundamental model of disjoining Deryagin pressure adequately describes the WRC section with the dominance of surface water-retention forces reflected by the exponential dependence of energy (potential) and the thickness of the water film (*h*, [m]) or the mass content of water (*W* [kg/kg]) [34,35]:(5)$$\left|\Psi \right|=a\cdot exp\left(-\frac{h}{\lambda }\right)=a\cdot exp\left(-bW\right),\;b=\frac{1}{S\rho_{l}\lambda }$$
where *a* [J/kg] is a physically based parameter reflecting the surface shape and potential (charge); λ [m] is the length of correlation for the structural forces or effective Debye thickness of the double electric layer for ion-electrostatic forces; *S* [m^2^/g] is the variable specific surface of the interphase boundary; and *b* [kg/kg] is the physically based parameter controlled by λ and *S* according to the modern concept of thermodynamics of water retention and dispersity in soils [35]. In a standard stable state with a minimum water film thickness (*h*_st_ = 2λ_st_), the specific surface area (*S*_st_) is defined by the slope of the WRC, as follows [34,35]:(6)$$S_{\mathrm{st}\;}=\frac{1}{br_{0}\rho_{l}}\sqrt{2\exp(-2)}\approx \frac{1}{2br_{0}\rho_{l}}$$
where *r*_0_ = 1.38 × 10^−10^ m is the crystallochemical radius of a water molecule. For comparative purposes, we used the ratio γ_1_ = *S*_st_/*S*_st_^0^, showing how much the dispersity of the soil substrate (*S*_st_^0^ [m^2^/g]) increased under the influence of the hydrogel.

The fundamental Model (5) was also used to estimate the total energy of water retention by the solid phase of the sample (*E*_t_ [J/kg]) [34]:(7)$$E_{t}=\int_{W=0}^{W\to \infty }a\cdot \exp(-bW)=a/b;$$
the generalized Hamaker constant (*A*_G_ [J]), characterizing the molecular (dispersive) component of the disjoining pressure [34,35]:(8)$$A_{G}=\frac{24\alpha \pi r_{0}\rho_{l}}{\sqrt{2\exp(-2)}}\approx 48a\pi r_{0}\rho_{l};$$
the critical water content of the particle coagulation threshold in two-phased gel systems [34,35]:*W*_cr_ = 2/*b*;(9)
and the critical concentration corresponding to the coagulation threshold (*C*_cr_ [mol/L]) of electrolytes in the intermicellar solution [37]:(10)$$C_{\mathrm{cr}}\approx 213\frac{{\left(RT\right)}^{5}{\left(\xi_{0}\xi \right)}^{3}}{A_{G}{}^{2}{\left(Fz\right)}^{6}}$$
where *F* [C/mol] is the Faraday number; *z* is the ion valence; *ξ* [F/m] is the electric constant; *ξ* [dimensionless] is the dielectric permittivity of the dispersion medium (water).

The product *W*_cr_ × *C*_cr_, [mmol/kg] represents the critical concentration of the electrolyte causing the total collapse of the gel structure, relative to the dry mass of the polymer or soil substrate. By analogy with dispersity, for a comparative assessment it is convenient to use the dimensionless ratios γ_2_ = *E*_t_/*E*_t_^0^ and γ_3_ = (*A*_G_^0^/*A*_G_)^2^ characterizing the increase in the surface energy of the soil solid phase and the resistance of colloidal particles to coagulation (according to criterion (10)) under the influence of gel-forming soil conditioners. Here and below, the superscript «0» signifies the value of the indicator for the original soil substrate (untreated control).

The temperature of the «water–ice» phase transition (*T*_F_, [K or °C]), depending on the degree of swelling of the gel, or the corresponding water content in the soil sample, was determined directly using the cryoscopic method [35] with DS1923 loggers (Maxim Integrated Products, Inc., Wilmington, MA, USA) or calculated from the thermodynamic potential of water, according to the equation [34]:|Ψ|= *L*·(*T*_F_ − *T*_0_)/(*T*_0_*M*)(11)
where *L* = 6013 J/mol is the latent freezing heat for water, *T*_0_ = 273 K is the freezing point of pure water, and *M* [kg/mol] is the molar mass of water. A freezing temperature near 1 °C corresponds to the conditional limit of water available for plants (potential 1224 J/kg, which is close to the critical root potential of 1500 J/kg, according to [38]).

Additional calculations included WRC-assessment of agronomic indicators of total water capacity (*W*_s_), field capacity (*FC*) and the range of soil moisture available to plants (*AWR*) [34]. The total water capacity or water content in the state of soil saturation was estimated using the van Genuchten model at |Ψ| = |*P*| = 0. The field capacity was determined by the Voronin method [34,39] as the WRC crossover point with the curve lg|Ψ| = 1.17 + *W*. The *AWR* indicator was calculated as the difference between the field capacity and the wilt point (*WP*): *AWR* = *FC* − *WP*. The wilt point of plants according to Richards-Weaver [38] is the water content at the absolute value of soil water potential |Ψ| = 1500 J/kg.

#### 2.4.3. Soil Hydraulic Properties

Along with water retention, the water-conducting properties are of paramount importance in determining the water regime of the soil, root water consumption, and plant productivity, in accordance with modern computer models of energy and mass transfer in a soil–plant–atmosphere system [40,41]. The basic indicator of saturated hydraulic conductivity (*K*_0_ [m/s]) was estimated using the variable head method in an experiment preceding the equilibrium centrifugation. Centrifuge tubes with water-saturated samples were extended with plastic cylindrical nozzles of the same diameter. These nozzles were filled with water up to the top. By periodically measuring the water level in the nozzles above the soil surface (*h*(*t*)) and approximating the experimental data with an exponential relaxation model with constant empirical parameters *a* [m] and *b* [s^−1^]:*h*(*t*) = *a·*(1 − exp(−*b·t*)),(12)
it was easy to obtain a saturated hydraulic conductivity indicator:*K*_0_ = *L·b*,(13)
where *L* [m] is the height of the soil sample in the centrifuge tube. For a comparative assessment, it was convenient to use the *K*_0S_/*K*_0_ ratio of water conductivity in the initial soil substrate (*K*_0S_) to a similar value for samples treated with gel-forming soil conditioners (*K*_0_).

Unsaturated hydraulic conductivity as a function of the potential (pressure) of water in a porous medium (*K*(*P*) [m/s]) was estimated synchronously with the water potential, taking into account the drainage rate of soil samples in the centrifuge experiments [31]. For this purpose, the weight loss of the sample and the corresponding water content at each stage of centrifuge rotation (*W*(*t*)) was measured periodically over time. The experimental data (*W*(*t*)) were approximated using an exponential relaxation model:*W*(*t*) = *W*_e_ + (*W*_0_ − *W*_e_)·exp(−*k*·*t*)(14)
where *W*_0_ and *W*_e_ [kg/kg] are the initial (before centrifugation) and final (equilibrium at a given rotation speed) water content; and *k* [s^−1^] is the empirical constant of water content relaxation to the equilibrium state. The *k* value was used in a modified Darcy equation to calculate the unsaturated hydraulic conductivity at a given centrifugation step:(15)$$K\left(P\right)=\frac{k\left(W_{0}-W_{e}\right)L^{2}\rho_{b}g}{\left(P_{0}-P_{e}\right)}$$
where *P*_0_ and *P*_e_ [Pa] are the initial and equilibrium absolute pressure of soil water at a given centrifugation step; *L* [m] is the height of a sample; ρ_b_ [kg/m^3^] is the soil bulk density, and *g* [m/s^2^] is the acceleration of gravity.

#### 2.4.4. Evaporation Rate

This important technological indicator (*Q* [kg/(kg·day)], [mm/day]) was estimated using the weight method from the mass loss (Δ*m*) of a wet sample in a glass with a known cross-sectional area (*S*_i,_ [m^2^]) followed by division by the mass of the dry sample or by the area of the evaporator:*Q*_1_, [kg/(kg·day)] = Δ*m*/(Δ*t·m*_0_) = Δ*W*/Δ*t*(16)
*Q*_2_, [mm/day] = 1000·Δ*m*/(Δ*t·S*_i·_ρ_ℓ_)(17)
where *t* [day] is the time of the experiment, and *m*_0_ [kg] is the mass of an absolutely dry sample. As a comparative control, the evaporation of pure water (*Q*_0W_) at a fixed temperature and air humidity was used. In this case, a convenient indicator of the effectiveness of the soil gel-forming conditioners was the ratio *Q*_0W_*/Q* averaged over the first five days of the experiment. For soil samples, it is also convenient to use the *Q*_0S_*/Q* ratio, which shows the intensity of evaporation reduction under the influence of hydrogels relative to the untreated control (*Q*_0S_).

#### 2.4.5. Intensity and Degree of Swelling

Both indicators were determined in transparent graduated plastic or glass tubes with a known cross-sectional area, and with capillary feeding of the sample with water (or saline solutions) without external load (free swelling) or with a limiting load of a known mass (*m*_L_ [kg]). Sample height (*h*_S_ [m]) measurement with a micrometer was used to accurately estimate the volume of the swollen sample (*V*_S_ [m^3^] = *S*_i,_*h*_S_). The degree of swelling of the gel (*SD* [kg/kg]) was calculated as the mass of absorbed water to the mass of dry gel (*m*_0_ [kg]):

The external pressure (*P*_E_ [Pa]) was estimated using the weight of the load:*SD* = *V*_S·_ρ_ℓ_/*m*_0_(18)
*P*_E_ = *m*_L·_*g*/*S*_i_(19)

Swelling in distilled water was characterized using the *SD*_W_ index, reaching values of 500–1000 kg/kg and more in acrylic superabsorbents. However, most known gel conditioners drastically reduce swelling under osmotic stress (osmotic pressure) and/or external mechanical pressure acting in real soils. Such a limitation is conveniently expressed as a reduction in *SD* relative to the standard degree of swelling. If we take *SD*_W_ = 400 kg/kg as a standard (transfer to gel of all water in sandy soil with total water capacity *W*_s_ = 40% at a working 0.1% dose of gel), then the gel resistance to osmotic pressure, expressed as a percentage of the standard, will be *SD*_S_/4, where *SD*_S_ [kg/kg] is the degree of swelling in an aqueous solution of potassium chloride with a concentration of 2 g/L. This concentration, according to the van’t Hoff equation, creates an osmotic pressure close to one atmosphere (123 kPa with an isotonic coefficient of 1.88 at room temperature 293 K). According to Formula (11), this corresponds to a freezing point of about 0.1 °C. By analogy, to assess the availability of water to plants, it is possible to use the indicator Δ*SD*_1_ = (400 − *SD*_1_)/4, where *SD*_1_ is the amount of water (degree of swelling) in the gel, freezing at a temperature of 1 °C.

#### 2.4.6. Strength of Samples Treated with Gel-Forming Soil Conditioners

To assess strength, soil aggregates of known diameter (*D*_A_ [m]), usually 3–5 mm, were crushed with a finger on a platform of technical scales. The indications of the scales (*m*_A_ [kg]) at the moment of destruction of the aggregate, easily perceptible tactilely, make it possible to evaluate the strength (destruction pressure, *P*_A_ [Pa]):*P*_A_ = *m*_A_·*g*/(π *D*_A_^2^/4)(20)

#### 2.4.7. Biological Activity and Resistance to Biodegradation

The laboratory indicator of microbial biological activity of samples is their basal respiration (*U*_m_ [mg/(kg·h)]), estimated using CO_2_ emission or O_2_ absorption by the mass of the solid phase of the test sample (*m*_s_ [kg]) [30,42,43]. To obtain this, the samples were incubated in hermetically–sealed vials at certain levels of temperature and humidity (it is convenient to combine this experiment with centrifugation for a joint assessment of the potential and *U*_m_, depending on the soil water content). For detection of CO_2_ and O_2_−, gas chromatograph or infrared and electrochemical gas analyzers can be used. We used a PKU-4HP CO_2_ infrared analyzer (EKSIS JSC, Moscow, Russia, https://www.eksis.ru; accessed on 24 October 2022). When determining the basal respiration by CO_2_ emission, a modified method [30] was used, taking into account the interfacial interactions of CO_2_ in the soil (sorption, dissolution) by means of rapid thermo–desorption removal of immobilized CO_2_ in a microwave oven after standard incubation. The calculation of basal respiration using data of the increase (CO_2_) of the gas volume content (Δ*X*) during the incubation time (Δ*t*) was carried out according to the formula:(21)$$U_{m}=\frac{P_{g}MV_{g}\Delta X_{\%}}{10^{2}RTm_{s}\Delta t}=\frac{P_{g}MV_{g}\Delta X_{\mathrm{ppm}}}{10^{6}RTm_{s}\Delta t}$$
where *R* is the universal gas constant (8.314 J/(mol·K); *T* [K] is the absolute temperature; *P*_g_ [Pa] is the barometric gas (atmospheric) pressure; *M* [kg/mol] is the molar mass of CO_2_; *V_g_* [m^3^] is a volume of the gas phase in the incubation vial; and % or ppm are the usual units of the volume content of gases for different gas analyzers.

Information about basal respiration, expressed in [mgCO_2_/(kg·h)], as well as the mass fraction of organic carbon in the sample (*C*_org_), expressed in [%], allows us to evaluate the resistance of organic material to biodegradation in the form of a half-life index (*T*_0.5_ [yr]), according to [30]:
(22)$$T_{0.5}=\ln(2)/k_{0},\;\;\;k_{0}=\frac{T_{b}}{T_{0}}\left(\ln100-\ln\left(100-24\cdot 10^{-2}\frac{12U_{m}}{44C_{\mathrm{org}}}\right)\right)$$
where *T*_0_ is the time scale for the process of biodegradation in yrs, 12/44 is the ratio of the molar masses of carbon and CO_2_, 24 × 10^−2^ is the conversion factor from hours to days, from milligrams to kilograms and from % to kg/kg; and *T_b_* is the average yearly period of biological activity, expressed in days.

#### 2.4.8. Biological Activity against Pathogenic Organisms

The use of gel structures as carriers for biocides (fungicides) for soil poses the problem of the quantitative laboratory assessment of their biocidal activity against selected pathogens. This methodological task was solved in our work [3] using the classical method of seeding pathogen isolates on nutrient media in Petri dishes (radial growth from the center of the dish) containing protective gel structures with different doses of fungicides. During incubation at the optimum temperature (25 °C), the dishes were periodically photographed to fix the area of the growing colonies of pathogens (*S*) with further digitization using graphic analysis in the computer program Adobe Photoshop CS2. An estimation of the area was carried out in three replications for the control and each experimental variant with different concentrations of fungicides in all types of hydrogel. A comparative measurement of the areas was performed when the colony sizes in the control reached the total area of the Petri dish (*S*_P_). The dimensionless *x* = (*S* − *S*_0_)/(*S*_P_ − *S*_0_) indicator adjusted for the inoculation area (*S*_0_) was used to calculate the median or half-maximal (50%)effective concentration (*EC*_50_) of the fungicide, when *x* was equal to 0.5, as well as the effective concentration (*EC*_95_) of the complete (95%) growth inhibition, when *x* = 0.05, on the basis of the exponential model [3]:*C*_f_ = *A*·exp(−*f*·*x*)(23)
where *C*_f_ is the concentration of the fungicide [ppm]; *A* [ppm]; and *f* [dimensionless] are the empirical parameters of the model. The corresponding formulas for the calculation are as follows:*EC*_50_ = *A*·exp(−0.5 *f*)(24)
*EC*_95_ = *A*·exp(−0.05 *f*)(25)

The protective fungicidal properties of the hydrogel compositions were tested using isolates of late blight (*Oomycete Phytophthora infestans* (*Mont.*) *de Bary*) obtained from the State collection of phytopathogenic microorganisms at the All-Russian Research Institute of Phytopathology (http://vniif.ru/vniif/structure/collection; accessed on 24 October 2022). We used collection strains isolated in 2003–2007 from the Red Scarlett and the Sante potato varieties in the Moscow and Krasnodar regions of the Russian Federation. To identify the strains, a set of differentiator varieties including 22 genotypes from R1 (CIP N800986) to R11 (CIP N800996) obtained from the International Potato Center (CIP, Peru, https://cipotato.org; accessed on 20 October 2022) was used before the isolate was placed in the collection. The control experiment was carried out on a potato-dextrose agar medium (PDA) for the growth of pathogenic microorganisms (grated potato 200 g; dextrose 20 g; agar 20 g; tap water 1000 mL, pH 6.5–7.0). In testing anti-pathogenic gel structures, the hydrogels and agar were mixed in a 1:1 ratio with a dose of 10 g of each 

Substance per 1 L of medium. The inoculation was carried out in the center of a 100 mm diameter Petri dish, in the form of an oomycete mycelium excision 5 × 5 mm.

#### 2.4.9. Guidelines for the Main Quality Indicators

A comparative assessment of the various gel-forming soil conditioners and their compositions with coarse-textured soil substrates was based on simple approximate gradations («lack-norm-excess») of the main quality indicators discussed above with the necessary technological and environmental comments (Table 2 and Table 3).

### 2.5. Other Laboratory Methods; Experimental Data Processing

The particle size distribution of the samples was determined by laser diffractometry [44] on a Mastersize 3000 Malvern, UK instrument. The content of organic carbon (*C*_org_, [%]) in the samples after the destruction of carbonates by hydrochloric acid was determined by coulometric titration using the AN-7529 analyzer (Russia). The electrical conductivity and pH in the filtrates from water-saturated samples were analyzed with a HI 98130 Combo instrument (HANNA Instrument). Approximation of experimental data using Models (5), (12), (14) and (23), etc. was carried out in the S-Plot 11 program using the Regression Wizard nonlinear regression package. Statistical processing was carried out in MS Excel 16 spreadsheets and in the R (3.5.3) program.

## 3. Results

### 3.1. Composition and Technological Properties of Polymer Hydrogels

#### 3.1.1. Polymeric Matrix Composition and Swelling

Depending on the soil conditions, water quality and selected type of vegetation, the composition and ratios of components in composite soil conditioners can be varied to obtain optimal technological characteristics. For example, changing the ratio of copolymers of acrylamide and acrylates can be used to obtain materials capable of swelling in quite saline solutions (Figure 1A). The concentration of potassium chloride 3 g/L inhibits the swelling of the gel at a 10/90 ratio of acrylamide and acrylic acid salts by 10 times and up to 20 times in the case of equal proportions of copolymers. At lower salt concentrations, osmotic stress remains a powerful limiting factor in the swelling of well-known brands of hydrogels such as Aquasorb (SNF-group, https://www.snf-group.com; accessed on 20 October 2022) or Zeba (UPL, https://www.upl-ltd.com; accessed on 20 October 2022) based on PAA, acrylic acid and starch, explaining the failure of these materials in arid irrigated agriculture with saline soils and water (Figure 1B). An increase in the proportion of acrylates and the introduction of silver ions in Aquapastus composite materials leads to a higher salt resistance and higher water absorption compared to the known analogues (Figure 1B). The degree of swelling depending on the electrolyte concentration (*C*_E_ [g/L] is described using a simple relaxation model:*SD* = *SD*_r_ + *SD*_W_·exp(−*C*_E_/*C*_sf_)(26)
where *SD*_W_ is the maximum degree of swelling in pure water; *SD*_r_ is the residual degree of swelling after salt depression; and *C*_sf_ [g/L] is the concentration scale factor. For potassium chloride concentrations from 1 to 3 g/L, the values of parameters Equation (26) gave the following ranges of values: *SD*_W_ from 445 ± 27 kg/kg (Zeba) to 1113 ± 29 kg/kg (A22Ag); *SD*_r_ from 94 ± 15 kg/kg (Zeba) to 188 ± 16 kg/kg (A22Ag); *C*_sf_ from 0.21 ± 0.01 g/L (Aquasorb) to 0.33 ± 0.06 g/L (Zeba), statistically significant at *p*-value 0.0001–0.03.

Since the total water capacity (porosity) in the sands rarely exceeds 40% (mass), loose-crosslinked polymers with a swelling degree of 1000 kg/kg will not be able to realize this in a rigid pore space at cost-effective technological doses of about 0.1%. The maximum amount of water in such soils available for binding into a gel structure will not exceed the swelling degree of 40/0.1 = 400 kg/kg. Reducing the dose of gel to realize the maximum degree of swelling, of the order of 1000 kg/kg is impractical, since the strongly swollen gel loses its shape and changes to a viscous flow in the pores of the soil (leaching).

Therefore, for gel-forming soil conditioners, it is better to reduce the degree of maximum swelling to 500–600 kg/kg; for example, by increasing the dose of a crosslinking agent (N,N′-methylene bisacrylamide for Aquapastus gels). To further strengthen the polymer matrix, reduce the cost of materials and give them the amphiphilic properties necessary to retain organic pesticides and structure the soil, we used various biopolymer fillers (Figure 1C,D). Fillers in the form of waste from the biocatalytic production of PAA and dispersed peat practically do not change the degree of swelling of acrylic gels in doses up to 25%, after which it begins to decrease. Figure 1D reveals a rather wide area of variation in the concentrations of copolymers and peat filler, where the degree of swelling remains in the optimal technological range of 500–600 kg/kg. 

The swelling of hydrogels in the pores of the soil may not only not be realized because of limiting physical and chemical factors. The time to establish equilibrium is also important here. Often, the dry gel simply does not have sufficient time to retain water, which quickly infiltrates into sandy soils. Kinetic curves of limited swelling (external pressure near 6 kPa) show the rather slow achievement of equilibrium over a day or more (Figure 2). We propose a simple relaxation model to describe this process:*SD*(*t*) = *SD*_0_ + *SD*_L_ (1 − exp(−*m*_L_*t*)(27)
where *SD*_0_ and *SD*_L_ are the initial and final (limiting) degree of swelling of the hydrogel, *m*_L_, [h^−1^] is the kinetic constant, approximately (if *SD*_0_ << *SD*_L_) related to the half-life of the swelling process: *T*_0.5_ ≈ ln(2)/*m*_L_. The analysis of the parameters presented in Figure 2 shows that the half-life of the swelling process varied from 5.5 ± 0.1 1 h (Aquasorb) to 8.3 ± 0.2 h (A11). 

#### 3.1.2. Water Retention, Dispersity and Stability of Gels with Regards to Physical and Chemical Factors

A thermodynamic assessment of the water retention in a wide range of water potentials from 30 J/kg to 240 kJ/kg indicated the similar nature of the dependence of the potential on water content for all samples of hydrogels (Figure 3A). This dependence can be described adequately using the well-known Campbell-Brooks-Coury power-law hydrophysical model [40,41]: |Ψ| = Ψ_0_*W^−n^*, where Ψ_0_ and *n* are empirical parameters. The proximity of the parameter *n* to 1 probably indicates the osmotic mechanism of water retention in the gel structures, since the concentration of the reticular polymer is inversely proportional to the water content in it (the degree of swelling). The calculated values of *SD*_S_/4 and Δ*SD*_1_ revealed the different qualities of the compared gel samples (Figure 3B, Table 2). The hydrogel Zeba had the least resistance to pressure and freezing (*SD*_S_/4 = 20.3 ± 1.8, *SD*_1_/4 = 98.5 ± 0.3); the best indicators *SD*_S_/4 = 42.5 ± 3.8, *SD*_1_/4 = 96.7 ± 0.3) were obtained for the new A22Ag composite material with a peat filler and ionic silver.

Figure 4A,B represent the sorption part of the water retention curve and the thermodynamic indicators of the quality of the *S*_st_, *E*_t_, and *W*_cr_ gels calculated from it using the fundamental Model (6).

The exponential Model (6) adequately describes this part of the gel’s WRCs (determination coefficient R^2^ = 0.996–0.998) in a wide range of absolute values of the water potential from 2734 J/kg (equilibrium with relative humidity of 98%) to 894,533 J/kg (water potential for a standard drying temperature of 105 °C, according to Equation (3)). On a semi-logarithmic scale, the lines |Ψ|(*W*) fan out from a common point with a conditionally zero water content in accordance with the dispersity, as predicted by the fundamental model of the disjoining pressure—Equation (6) [34,35]. The minimum dispersity was obtained for the Zeba hydrogel (*S*_st_ = 225 ± 31 m^2^/g), and the maximum was for the new A22Ag material (*S*_st_ = 879 ± 29 m^2^/g). Aquasorb and the A11 hydrogel had similar dispersity characteristics (*S*_st_ = 627 ± 22 m^2^/g and 603 ± 25 m^2^/g, respectively), so their WRCs were also similar. The total water retention energy reached its maximum value (*E*_t_ = 173 ± 12 kJ/kg) for the A22Ag gel. For the Aquasorb and A11 gels, this indicator did not differ statistically significantly (*E*_t_ = 137 ± 10 kJ/kg and 126 ± 9 kJ/kg, respectively). The minimum water-retention energy (60 ± 4 kJ/kg) was found in the Zeba material. The *W*_cr_ index changed similarly from the minimum value of 12 ± 0.2% for the Zeba hydrogel to the maximum value of 47 ± 0.7% for the new material A22Ag.

The limiting concentration of the monovalent binary electrolyte causing the collapse of the gel (*C*_cr_) varied from 53 ± 3 mmol/L in the Zeba material to 96 ± 7 mmol/L in the new hydrogel A22Ag. In the case of sodium chloride with a molar mass of 58.5 g/mol, the critical concentrations of complete collapse of gel structures were 4.6; 5.0; 5.6; 3.1 g/L in the compared materials: Aquasorb; A11; A22Ag; Zeba, respectively. Hence, for widespread sodium chloride (marine type) salinization of soils, the effectiveness of such materials will be practically zero if the soil solution and/or irrigation water contains more than 3–5 g/L of salts. According to the well-known classification of salinity in soils [45], this concentration of soil solution corresponds to the category «slightly saline soils» with an electrical conductivity from 4 to 8 dS/m. In saline (EC = 8–16 dS/m) and highly saline (EC > 16 dS/m) soils, the application of hydrogels will thus be generally ineffective. In the case of two and three valence ions, the critical concentration of the loss of stability of gel structures decreases sharply in proportion to the valence of the ion by 6 degrees in accordance with the well-known Schultz–Hardy rule, reflected in Formula (10). These patterns explain the potential risk of using superabsorbents in arid climatic conditions with saline soils and pose the challenge of synthesizing gel structures that are more resistant to salinity.

Our new Aquapastus materials demonstrate a higher resistance to the salinity factor than the well-known brands Aquasorb and Zeba in the range of relatively small electrolyte concentrations up to 3 g/L (Figure 1B). However, the salinity level of 5–5.6 g/L was critical for them, which is a slightly higher limit than for the known analogues (3.1–4.6 g/L). The Hamaker constants change rather weakly in a number of the compared polymeric materials: 3.3 × 10^−19^ J (Aquasorb); 3.1 × 10^−19^ J (A11); 2.9 × 10^−19^ J (A22Ag); 4.0 × 10^−19^ J (Zeba). Their ratio (criterion γ_3_) shows that the potential resistance to coagulation of the most promising A22Ag material was only 1.2 times higher than that of Aquasorb. A more detailed criterion for the coagulation threshold *C*_cr_ × *W*_cr_ showed the lowest resistance to electrolytes for the Zeba material (6 mmol/kg), and the maximum resistance for the A22Ag hydrogel (45 mmol/kg), while the Aquasorb and the A11 materials were characterized by the norm for this indicator (26 and 27 mmol/kg), according to the Table 2. Generally, in accordance with the information in Table 2, the combined PAA-starch material Zeba had lower indicators for the degree of swelling, dispersity, total water-retention energy, resistance to pressure and coagulation; while the Aquasorb and the A11 hydrogels were mainly characterized by normal values for these indicators, and the innovative material A22Ag often demonstrates the best values for the quality indicators exceeding the norm.

#### 3.1.3. Biodegradation of Hydrogels and Its Inhibition

Along with resistance to physical and chemical factors, an important technological indicator of gel-forming soil conditioners is their resistance to biodegradation. A priori, acrylic hydrogels should be susceptible to microbial degradation, since their C/N ≈ 2.5 ratio is significantly less than the conditional C/N ≤ 20 boundary, which denotes a high biodegradability of plant residues in the soil [46]. Experiments on the basal respiration of hydrogels confirmed this assumption (Figure 5A,B) [30]. The magnitude of CO_2_ emission of pure hydrogels ranged from 28 ± 7.2 to 280 ± 72 mgCO_2_/(kg∙h), which meant high and very high biological activities, in accordance with the known criteria for soils [6,42,43]. This activity is sufficient to cause considerable damage to synthetic gel structures in the soil. The half-time period of microbial decay for the different gels ranged from 0.5 to 2.6 years and only in the sample A22 with amphiphilic fillers in the form of dispersed peat was 5.2 ± 1.5 years.

The use of silver inhibitors significantly reduces the respiration and biodegradation in pure gel–silver compositions (Figure 5A,B). In the case of silver ions at a dose of 10 ppm, the *U_m_* decreases 13–30 times except for the Aquasorb gel, where *U_m_* was reduced to no more than three times. Doses of 100–1000 ppm inhibited *U_m_* 20–60 times without significant differences in the effects between them. As a result, the calculated values for the effective half-lives *T*_0.5_ increased to 5–30 years in gel structures with 10 ppm Ag and up to 25–50 years when the content of silver ions ranges from 100 to 1000 ppm. Silver nanoparticles had the same or a somewhat greater inhibitory effect. At a dose of 10 ppm Ag, the value of *U_m_* is equal 6–25 mg CO_2_/(kg∙h) and the *T*_0.5_ varied from 6 to 20 years, and at doses of 100–1000 ppm the *U_m_* was equal 1.8–2 mg CO_2_/(kg∙h) or *T*_0.5_ = 24–105 years.

#### 3.1.4. Laboratory Analysis of SGS Protective Antimicrobial Properties

In addition to the resistance to biodegradation, the incorporation of biocides into gel structures is essential for the controlled release of anti-pathogenic gel systems. Figure 6 illustrates the visual antifungal effect of the composition A22 with silver nanoparticles, as well as our new method for *EC*_50_ and *EC*_95_ estimation using the exponential Model (23) [3]. The estimated *EC*_50_ and *EC*_95_ values for different hydrogels are shown in Table 4.

The EC_50_ of the studied gels varied within a range of 0.9 ± 0.1 to 63.0 ± 19.6 ppm. Gel compositions based on silver nanoparticles had lower-range average values (1–17 ppm). Similar compositions with ionic silver and synthetic fungicide based on Azoxystrobin were characterized by higher EC_50_ values of 17–60 ppm, i.e., a lower fungicidal effect. The type of hydrogel (hydrophilic—Aquasorb, A11 or with amphiphilic filler—A22) may have affected the fungicidal properties, which was confirmed using the LSD method at *p* = 0.05 significance levels. The complete inhibition of pathogenic agent activity EC_95_ ranged from 187 ± 13 to 448 ± 31 ppm. These values exceeded the estimations of the fungicidal efficiency 10–50 times or more based on the EC_50_ alone.

### 3.2. Laboratory Testing of Soil–Gel Compositions

#### 3.2.1. Thermodynamic Assessment of Water Retention and Dispersity

The water-retention characteristics of the tested hydrogels and the structural curves of pore size distribution calculated by WRCs are shown in Figure 7 (sandy substrates) and Figure 8 (loamy–sandy Arenosols). The continuous lines in the main figures of the WRCs represent the fundamental ion–electrostatic model of disjoining Pressure (5) and the standard van–Genuchten model [33]. The dotted line is the secant for determining the field water capacity using the Voronin [34,39] method. All obtained data of the WRCs fitted adequately (determination coefficient R^2^ = 0.992–0.999, relative standard approximation errors ≤ 1.5–2%) using the van Genuchten model [33] in the capillary range (|Ψ| = 0–1000 J/kg) and using the fundamental ion–electrostatic Model (5) in the range of surface absorption mechanisms (|Ψ| > 1000 J/kg). The selective parameters of the van Genuchten model and their statistics are contained in Table 5 and Table 6.

All tested soil–gel compositions were characterized by a stable increase in water-retention proportional to a dose of the hydrogels over a wide range 0–10^6^ J/kg of the absolute soil water potential values. The WRCs of the tested compositions were often shifted to the right with respect to the control position (mineral soil substrates), which indicates an increase in the energy of the water retention and water capacity of the samples. Alternatively, the pore size spectra had a tendency to move left, towards smaller sizes. This tendency most likely reflects the effect of aggregation of the mineral mass by the gel-forming soil conditioners. Visual analysis of the WRCs confirmed the maximum effect of the gel–forming conditioners in the sands. Here, even the smallest dose of 0.1% statistically significantly shifted the WRCs to a region of higher water retention energy and water capacity (Figure 7). In loamy–sandy Arenosols, the effect of the hydrogels was less, and a higher dose of 0.3% gel was required to significantly improve the water retention (Figure 7). This was presumably due to both the tighter pore space being filled with silt particles and fine sand (Table 1), and the osmotic stress from the highly soluble salts presented in the carbonate Arenosol from the Persian Gulf, with a minimal effect from the hydrogels.

Table 5 and Table 6 represent the traditional agrophysical indicators of field water capacity, wilting point, the available soil water range the total water-retention energy, generalized Hamaker constants and the dispersity parameter of a specific surface area estimated using the slope of WRC. For sandy soil substrates, the field water capacity at hydrogel doses of 0.2–0.3% from the soil dry mass reached 15–29% compared to 3.0–4.6% in the untreated control, which means that the water retention in such doses increased 5–10 times corresponding to a translation of the original sandy substrates to a loamy soil with this indicator. At the same time, the increase in the wilting point rarely exceeded 2–5 times. As a result, the *AWR* was expanded 4–8 (up 10–13) times compared to the original mineral substrates (Table 5). Similar results of a strong increase in water retention from small doses of acrylic polymer hydrogels in sandy soils were obtained in [13,14,15,16,17,18,19,20]. The effect of gel–forming soil conditioners in loamy–sandy Arenosols was significantly less. Here, the field capacity increased by no more than 1.7 times, and the *AWR* by no more than 1.5 times at a maximum dose of hydrogels of 0.3% (Table 6).

The specific surface area of the studied soil–gel compositions increased 2.5–3.8 times at hydrogels concentrations of 0.1% and up to 6.0–10 times at higher doses of 0.2–0.3%, reaching values of 40–60 m^2^/g (up to 73 m^2^/g), typical for loams. All the tested hydrogels increase the specific surface by more than 2 times and the total energy of water retention by more than 1.5 times; that is, effectively and very effectively, according to the gradations of Table 3 (criteria (γ_1_ and γ_2_). The aggregate stability index γ_3_ was near to the norm or exceeded it for most of the compositions, excluding materials with organic fungicide (A11Az and A22Qv), where this indicator showed a weak aggregate stability and the potential risk of gel collapse under salinity. Apparently, the organic fungicide increases the surface energy of the composition and, consequently, the mutual attraction of dispersed particles, which was confirmed by the maximum values of *E*_t_ (up to 14,737 J/kg) and generalized Hamaker constants (up to 3.3 × 10^−19^ J) in these samples. It is remarkable that increasing the specific surface area under the influence of hydrogels did not satisfy the additivity rule. Since the specific surface area of the gels themselves did not exceed 1000 m^2^/g (Figure 4B), the introduction of polymers into sandy substrates in doses of 0.1–0.3% should theoretically have increased their dispersity by no more than 1–3 m^2^/g, whereas the experimental evaluation gave significantly higher *S*_st_ values, up to 50–70 m^2^/g. Apparently, polymer gels cover sand particles with thin hydrophilic layers, significantly increasing the hygroscopicity and, consequently, the effective specific surface area as estimated by water sorption.

In general, from a technological point of view, the filling of the acrylic polymer matrix with various ingredients did not have a significantly negative impact on the water retention and dispersity of the studied soil–gel compositions [3]. The usage of amphiphilic fillers (humates, peat) in the production of the Ural Chemical Factory did not significantly change the water retention properties of the soil–gel compositions compared to hydrophilic formulations of A11 and Aquasorb (Table 5 and Table 6). A similar result, concerning the insignificant changes in water retention, was obtained by comparing the amphiphilic gels supplemented or not with electrolyte’s admixtures in the form of trace elements (A11HMZ vs. A11H). Inclusion of ionic groups of electrolytes in the acrylates and humates in the structure of the polymer matrix allowed us to solve the problem of the reducing of the hydrogel’s swelling, as affected by the osmotic stress [26]. Moreover, embedded cations can react during ion exchange and serve as a source of elements for plant’s mineral nutrition. As the experiment with the samples of A22Ag amphiphilic gel at a dose of 0.1–1% silver showed, the compositions based on this did not have significantly wors water retention, dispersity, and structural properties, which remained at the level of Aquasorb with the same concentrations of 0.1–0.3% (Figure 7A,C; Table 5).

#### 3.2.2. Saturated and Unsaturated Hydraulic Conductivity

Gel-forming soil conditioners have a strong effect on the water permeability of the soil [7,8,13,15] and this was also confirmed by our experiments (Figure 9 and Figure 10). Figure 9 shows the changes in the saturated hydraulic conductivity of monomineral sandy substrates (A) and polymineral loamy–sandy Arenosols from the Karakum desert and the Emirate of Dubai (B) under the influence of different doses of hydrogels and placement methods in the soil. 

All the studied gel-forming conditioners sharply (exponentially) reduced the saturated hydraulic conductivity in coarse soil substrates, with a significant effect observed starting from a minimum gel dose of 0.1%. This dose reduced the *K*_0_ in sands by 5–8 times or quite effectively, according to our guidelines in Table 3. For Arenosols with a lower initial *K*_0_ value, its decrease under the impact of a 0.1% hydrogel dose was not so effective and rarely exceeded 2 times. Higher doses of gel-forming soil conditioners reduced the saturated hydraulic conductivity by 20–140 times in sands and 17–60 times in loamy–sandy Arenosols, up to *K*_0_ = 2–4 cm/day, which is inherent in silty and clayey soils, according to [6]. All these results were obtained for mixtures of mineral soil substrates and pre-swollen gels (*SD*_W_ = 100 kg/kg). An alternative method is the layered placement of 100% gel without mixing with the soil. This also very effectively (60–80 times) reduces the saturated hydraulic conductivity of coarse-textured substrates and blocks the capillary resorption of water deep into the soil due to the formation of the so-called imperfect capillary barrier [47]. In general, the use of gel-forming soil conditioners greatly (up to 10–80 times) reduces the water permeability of coarse-textured soil substrates which, along with an increase in water retention, should minimize the unproductive water losses from the rhizosphere.

The effect of hydrogels on unsaturated hydraulic conductivity is more complex. The corresponding graphs and their approximation using the Campbell power model [40] are shown in Figure 10. They show the dual effects on the unsaturated hydraulic conductivity of the mineral sandy substrate. In the range of high soil water content and, consequently, low absolute values of soil water potential |Ψ| < 10–15 J/kg, there was a reduction of hydraulic conductivity up to 2–3 times at low gel doses of 0.05–0.1% and up to 10–50 times at higher concentrations of 0.1–0.2%. This was apparently due to the increase in the viscosity of the water solution and then to mechanical colmation by the swollen gel, as in the case of saturated flow (see above). With further draining of the sample (absolute water potential 20–700 J/kg), the conductivity values began, on the contrary, to increase in proportion to the dose of the polymer. Notably, the gel particles, localized in the sand, forming a network of thin water–conducting extra tracks in the structure of coarse-textured substrate with large pores, which cannot conduct the water at this pressure. The formation of thin macropores and mesopores (0.1–10 microns) under the influence of hydrogels was confirmed by the analysis of pore size distributions in the soil–gel compositions (Figure 10, inset). Dominant pores on the curves shifted to the left, toward the smaller pores, and the share of the total pore volume in the structure of the gel compositions also regularly increased with increasing doses of gels. Hence, the soil, conditioned by hydrogel, should had an increased hydraulic conductivity in the area of the capillaries and film water (|Ψ| > 10–30 J/kg up to 3030 J/kg) compared to the control sandy substrates, and this was actually observed (Figure 10). The presence of silver in the polymer matrix slightly reduced the effectiveness of the maximum dose of 0.3%. In these samples, the hydraulic conductivity and pore distribution became closer to the characteristics of pure gel structures with a lower dose of 0.2%. The power-law Campbell function [40], suitable for unsaturated flows, provided an excellent approximation of the actual data throughout the entire range of *K*(Ψ) variation, with determination coefficients R^2^ = 0.990–0.997 and statistically significant parameters of the approximation, which gradually reduced their values with increasing doses of polymers (Figure 10A,B).

#### 3.2.3. Evaporation of Water

An increase in water retention along with a decrease in hydraulic conductivity under the impact of hydrogels, explains their efficiency in preserving soil water and reducing its evaporation. Figure 11 shows the experimental results of an evaporation reduced by 2–3 times compared to pure water and 2–4 times compared to the untreated control in the form of water saturated quartz sand, which corresponds to the normal and high efficiency gel-forming soil conditioners, according to the gradations of Table 3. The intensity of the decrease in evaporation was directly dependent on the dose of the polymeric hydrogel. Filling the polymer matrix with dispersed peat (material A22) and the introduction of an organic fungicide slightly increased the suppressing effect for water evaporation compared to the pure hydrogel (A11), however, these changes are not statistically significant at the standard *p*–value = 0.01.

#### 3.2.4. Strength of Soil Aggregation

Particles of coarse textured soil substrates have a low surface energy and cannot spontaneously aggregate with each other. In nature, only the accumulation of fine-dispersed material (soil plasma) and cementing substances (carbonates, oxides and hydroxides of iron) in the process of soil evolution can lead to the formation of an aggregate structure in sands [6]. The lack of interparticle cohesion is the main reason for the strong wind erosion (deflation) of sands. The use of hydrogels, along with improving the water retention, hydraulic and surface properties of sandy soils, leads to the formation of soil aggregates, due to the gluing of sand particles with a gel after drying.

Figure 12 shows the increase in the strength of these aggregates from 20–40 to 100 kPa or more in the concentration range of acrylic hydrogels from 0.1 to 0.3%. Filling the polymer matrix with dispersed peat slightly increased the strength, possibly due to the reinforcement of the polymer network with finely dispersed peat particles, but these changes were not always statistically significant. A normal and high consolidation of sands with a strength of more than 50 kPa, according to the gradations of Table 3, requires doses of hydrogels of at least 0.2%.

#### 3.2.5. Basal Respiration and Resistance to Biodegradation

In contrast to pure gels, in the case of soil–gel compositions, we studied the response of basal respiration and, accordingly, biodegradability to the controlling factors of temperature (*T*, °C) and water content (dimensionless index *W*/*W*_s_, where *W*_s_ is the water content in the state of saturation of the soil). (Figure 13). An increase in temperature from 4 to 20 °C increased the basal respiration by 1.5–6.8 times, and from 20 to 30 °C—by 1.7–3.3 times. The influence of the temperature factor estimated for all variants of the experiment was adequately (R^2^ = 0.995, s = 0.04) described by the well-known function *Q*_10_, with the parameters of the temperature coefficient *Q*_10_ = 2.03 ± 0.13 (statistically significant at *p* < 0.04) and optimum temperature *T*_m_ = 30.1 ± 0.54 °C (statistically significant at *p* < 0.01). The dependence of respiration on water content had a more complex nonlinear form represented by the following function with an extremum in the range of 0.66–0.89 units of *W*/*W*_s_ [30]:*U*_(*W*)_ = *f*_(*W*)_*U*_max_(28)
(29)$$f_{\left(W\right)}={\left(\frac{W}{W_{m}}\right)}^{a}{\left(\frac{1-W}{1-W_{m}}\right)}^{b}\;$$
where *W*_m_ = *a/(a + b)* is the water point of the respiration extremum (*U*_max_) on the curve of the *U_(W)_* dependence, and *a*, *b* are empirical constants. The Model (28) adequately (R^2^ = 0.988–0.999, s = 0.02–0.05) described the experimental data with statistically significant at *p* = 0.0001–0.02 parameters *a* from 2.07 ± 0.16 to 8.69 ± 0.67 and *b* from 0.30 ± 0.06 to 2.53 ± 0.21.

As shown in Formula (22), the kinetic constants of biodegradation (*k*_0_) and half-life (*T*_0.5_) of the studied hydrogels were calculated based on the potential period of biological activity *T*_b_ = 200 days (Table 7). The constants *k*_0_ varied for compositions without silver inhibitors from 0.008 to 0.08 yr^−^^1^ in dry conditions (*W*/*W_s_* = 0.11–0.12), and from 0.25 to 2.35 yr^−^^1^ under optimal water content (*W*/*W_s_* = 0.71–0.89) in the temperature range from 4 to 30 °C. Similar indicators after using hydrogels with silver were significantly (up to 18–25 times) lower, they and varied in dried samples (*W*/*W_s_* = 0.14–0.17) from 0.0003 to 0.0045 yr^−^^1^, and under optimal conditions (*W*/*W_s_* = 0.66–0.88) from 0.009 to 0.135 yr^−^^1^ in the entire studied temperature range of 4–30 °C. Table 7 contains the values of *k*_0_ and the half-life for optimum humidity conditions at different temperature levels.

The biodegradable stability of the base gel-forming material A11 at high temperatures of 20–30 °C was small; within 1 year, more than half of the gel introduced into the soil had decomposed (*T*_0.5_ = 0.3–0.6 years). In cold (*T* = 4 °C) soil, the gel was preserved for several years (*T*_0.5_ = 2.3–2.8 years), but this situation cannot be permanent, even for a temperate climate, since low temperatures will only be found in winter (Table 7). At other times, coinciding with the growing season, heating will stimulate the biodegradation of the gels and reduce their effectiveness. However, the addition of 1% silver to the polymer matrix can significantly increase the stability of gel-forming conditioners. The example of the A22Ag material patented in the Russian Federation shows that even at the optimum temperature of 30 °C, which is typical only for soils in arid climates, the half-life of this composition in sandy substrates was prolonged to 5.1–5.5 years ((Table 7). At a cold temperature of 4 °C, the hydrogel could be preserved for 30 years or more (*T*_0.5_ = 27–76 years).

## 4. Discussion

### 4.1. Technological Properties of Gel-Forming Materials

#### 4.1.1. Swelling Degree and Kinetics

The results of a high (1000 g/g and more) degree of swelling in pure water for the studied composite hydrogels correspond to the known data for acrylic superabsorbents and their composites previously obtained [4,5,7,8,9,18]. The possibility of improving synthetic superabsorbents by incorporating organic (humates, lignohumates, peat, urea, amino acids, polysaccharides, etc.) and inorganic fillers (phyllosilicates, perlite, zeolites, thermal slags, etc.) was reported in [2,3,4,5,18]. Most fillers in small doses improve the swelling of composites in slightly salty solutions and under the influence of mechanical stress, creating additional ionogenic groups, and expanding and reinforcing the polymer mesh [3,5]. However, at high doses of fillers, for example, with polysaccharides, the swelling of composite superabsorbents decreases [5]. Apparently, this is explained by purely mechanical dilution, since biopolymer gels usually have a small degree of swelling (40–130 g/g) compared to synthetic acrylic superabsorbents [2]. Polysaccharide–based biopolymer hydrogels are more suitable for controlled release systems rather than water retention in soils where superabsorbents with a high degree of swelling (from 400 g/g and above) are required [2,3,4,5].

Osmotic collapse of superabsorbents, as one of the main factors reducing their effectiveness, has been described in many publications [4,18,20,26]. The polymer matrix and fillers contain ionogenic groups that create a total charge density along the chains during dissociation and increase the concentration of mobile ions in the gel. As a result, the internal osmotic pressure of the hydrogel increases, causing the movement of pure external water into the polymer mesh (swelling process). If the external water is saline and its osmotic pressure is higher than inside the polymer mesh, the hydrogel loses its ability to swell [18]. Our results (Figure 3A) indirectly confirmed the osmotic mechanism of swelling in proportion to the internal ionic concentration and, accordingly, inversely proportional to the water content (degree of swelling). In the external solution, not only the concentration, but also the charge of cations enhance the osmotic suppression of gel structures [4,48], and these facts are in full compliance with the theoretical stability criterion—Equation (10) for two-phase dispersed systems. Concerning the swelling capacity of composite acrylic superabsorbents in saline solutions, Shahid et al. [18] report materials with swelling degrees up to 140–390 g/g; i.e., exceeding our results. Unfortunately, the authors [18] did not analyze the concentration and chemical composition of “salt water”, so it is impossible to correctly compare these results with ours. The kinetics of swelling is no less important a technological factor of superabsorbents of water than the maximum degree of swelling. We obtained a good correspondence of the swelling kinetics of the relaxation Model (27) with the half-life parameters *T*_0.5_ = 5–8 h. Shahid et al. [18] reported similar time values for composite superabsorbents achieve maximum water absorption, from 5 to 14 h. A faster swelling of Aquazorb and other acrylic superabsorbents with half-life *T*_0.5_ = 0.5–1 h was observed by Shirinov and Jalilov [49]. Such different estimates can be explained by differences in the sizes of samples and their individual particles (fractions), research methods, and the presence or absence of load pressure, including the weight of hydrogels. The free, unrestricted swelling of fine hydrogel fractions does not reflect the real process in soils, where the pore space and pressure, as well as the speed and uniformity of water penetration can strongly suppress and delay swelling [49]. The usual speed of water movement for sands during the absorption of precipitation or irrigation water reaches 1–10 m/day or 4–40 cm/h [6]. Within 5–8 h, the water will leave the 0–20 cm rhizosphere layer with dry gel conditioners, which will not allow time to realize even half of their maximum degree of swelling. Therefore, gels should not be applied in a dry, but in a partially swollen state (the optimal degree of swelling is 100–200 kg/kg) in order to reduce infiltration (see Figure 9) and to retain water in the rhizosphere as much as possible. This technological method also makes it possible to evenly distribute the gel in the soil, since it is much easier to mix 1 part of wet gel with 10 parts of soil than 1 part of dry gel with 1000 parts of soil at a dose of 0.1% [3].

#### 4.1.2. Water Retention and Associated Quality Indicators

These water retention curves, with such a wide range of absolute values of water potential (pressure) from 30 to 240,000 J/kg (kPa) for gel-forming composite materials were probably obtained for the first time. The known methods for measuring the swelling pressure of hydrogels make it possible to obtain individual sections with a relatively low swelling pressure of up to 100 kPa [50,51], or, conversely, with a high pressure of 200–4200 kPa [52]. Combining all the data (Figure 3A), as already noted above, reveals the dominance of the osmotic mechanism of water absorption in the entire range of water potential, according to the Campbell–Brooks–Kuri model [40]. In the study in [50], a small external pressure near 10 kPa sharply reduces the degree of swelling for weakly cross–linked acrylic superabsorbents, up to 10–30 g/g. A further increase in pressure affected the water content of hydrogels less intensively. A pressure drop from 200 to 4200 kPa causes an increase in the polymer concentration from 0.03 to 0.3 g/g or a decrease in the degree of swelling from 33 to 3 g/g (3300–300% water content) [52]. These data are fairly consistent with the results obtained in our study (Figure 3A). Estimation of the specific surface area for hydrogels from 230–880 m^2^/g corresponds to the upper limit of the specific surface for finely–dispersed soils (150–400 m^2^/g), peat, humus, clay minerals and some food products (200–800 m^2^/g) [6,53,54]. The Hamaker constants for hydrogels are close to our estimates for soils and clay minerals [34,35] and somewhat higher than those for the soils in [55], which may be due to differences in the applied methodology and models.

#### 4.1.3. Biodegradation of Acrylic Composite Materials

The known published sources give conflicting information on this issue concerning mainly acrylamide and polyacrylamide (PAA). We excluded from consideration numerous advertising data about the stability of hydrogels in soils for 5–6 years or more (https://www.snf-group.com; accessed on 13 October 2022) because, in our opinion, they do not correspond with reality. Lentz et al. [29] reported a rather slow biodegradation of PAA, not more than 10% per year and primarily through the shear-induced chain scission and photodegradation. According to the standard exponential model [30], this corresponds to *k*_0_ = 0.11 yr^−1^ or the half-life of PAA hydrogels *T*_0.5_ = 6.6 years. This estimate significantly exceeds our results of *T*_0.5_ = 1–1.2 years for the radiation-crosslinked technical PAA obtained in our previous publication [29]. However, there are other data that offer an alternative perspective. Some studies found that the PAA and acrylamide monomer degradation in soil is very rapid, with half-life values of a few days [56,57,58,59,60]. Sojka and Entry [60] reported that PAA was completely degraded within 5 days after applying 0.05% to garden soil. Lande et al. [56] estimated the half-life of acrylamide monomer in agricultural soils as ranging from 18 to 100 h at a concentration of 25–500 mg kg^−1^ and a temperature of 20–22 °C. Soil microorganisms are capable of utilizing PAM or acrylamide as a source of nitrogen [57,59,60]. All these facts indicate the low potential stability of acrylic hydrogels in soils and the dominant mechanism of their biological (biochemical) degradation, rather than chemical or photochemical decomposition. Therefore, the introduction of inhibitors of biological activity into the composition of hydrogels should increase their resistance to biodegradation and our results clearly confirm this position (Figure 4) [3,30].

#### 4.1.4. Antimicrobial Properties of Composite Gel-Forming Materials

The active concentrations of 50% growth suppression for pathogens from biocides embedded in the composition of the hydrogels were in the range of 1–80 ppm. Similar results were obtained using the same laboratory’s microbiological method (silver ions or nanoparticles in the composition of dense, gel-like agar medium, or in the foams as saponins) in the following studies: *EC*_50_ from 10 to 50 ppm in [61,62,63]; diapason *EC*_50–_*EC*_99_ from 10 up to 111 ppm in [64]. The obtained *EC*_99_ values from 170 to 480 ppm were rather higher than the doses of 75–100 ppm of aquatic solution or dispersion of silver with a 99% suppressive effect reported in [65,66]. This may have been due to both, a slightly different mobility of fungicides in aqueous solutions and gels, and the specificity of the tested microflora in aquaculture.

As the Quadris synthetic fungicide has a half-life from 3 to 39 days (http://wineryclub.info/handbook/14-kvadris.html; accessed on 13 October 2022), this obviously guarantees its total biodegradation in our gel compositions, with a half–life of several years. In contrast to these biodegradable agent, the silver-gel compositions require an assessment of maximum allowable concentrations. For aquatic plants and aquaculture, a two-fold decrease in growth occurs at 10–20 ppm (up to 150 ppm for green algae) of silver ions or nanoparticles [67,68]. In the soil, the value *EC*_50_ for plants varies in the range of 50–1000 mgAg/kg of the soil solid phase [69]. In terms of the soil liquid phase, this gives a range from 250 to 5000 Ag ppm at 20% water content. A slightly smaller *EC*_50_ for ionic and colloidal silver ranging from 200 to 400 ppm, for soil earthworms was obtained after experiments with silver doses of 15 to 1000 mgAg/kg or from 75 to 5000 ppm, at 20% soil moisture, respectively [3,70]. Taking into account the obtained results, we incorporated 0.3–1.3% Ag (per mass) into the polymer matrix of our patented materials, A11-Ag and A22Ag. With a recommended dilution of 1:100 for the preparation of a fungicidal gel composition, this gave a concentration range from 30 to 130 ppm Ag, which is slightly more than the *EC*_50_ obtained in our experiments, but less than the threshold of 200–250 ppm that is dangerous for plants and earthworms in the soil.

### 4.2. Technological Properties of Soil-Gel Compositions

#### 4.2.1. Water Retention and Other Hydrophysical Properties

The thermodynamic assessment of water retention in soils under the influence of superabsorbents is quite common, however, as a rule, it is limited by the range of absolute values of the water potential (pressure) of 0–1500 kPa due to the use of suction–plate and pressure–plate equipment [7,10,11,12,13,14,17,20]. The disadvantage of these methods is the obligatory presence of porous membranes separating the free water and soil matrix. These membranes rapidly become clogged by hydrogel particles, which sharply reduces the rate of water removal, increases the time to establish thermodynamic equilibrium, and causes an apparent overestimation of the water-retaining capacity of gel–soil compositions, as well as a decrease in their hydraulic conductivity [13]. Our new methodology, based on a combination of centrifugation and thermo–desorption of water, successfully solves these problems and allows us to quickly obtain WRCs on a large number of samples with the necessary repeatability in the entire possible range of water potential from 0 (state of water saturation) to 0.8–1 million J/kg (state of conditionally “zero” water content after standard drying at 105 °C). The estimation of water retention for gel–soil compositions in our study was close to the data obtained using classical methods [7,10,11,12,13,14,17,20]. The water–retention energy and water capacity progressively increase in proportion to the dose of gel-forming soil conditioners, which was reflected by the shift of WRCs, as well as an increase in the percentage of water-retaining meso– and micropores (Figure 7 and Figure 8). Many researchers have reported a loosening of the soil and an increase in porosity (total water capacity) up to 1.5–2 times and field capacity up to 2–6 times under the action of hydrogels [7,8,10,11,15,17,19,20]. However, as Johnson [11] correctly notes, the field capacity does not always adequately reflect the conditioning effect of hydrogels. A more correct indicator is the *AWR*, which also estimates the potential increase in the percentage of water inaccessible to plants after the use of hydrogels in large doses. The *AWR* indicator in our study increased 4–8 times or more in sandy substrates and only 1.5 times in loamy–sandy Arenosol (Table 5 and Table 6), which is close to the data [11] for PAA hydrogels and [17,20] for acrylic composite superabstorbents. The most significant difference in our study was for the low optimal doses of gel–forming soil conditioners of 0.1–0.3%, sufficient to obtain these water retention results. Most publications reported higher optimal doses of superabsorbents (0.4–0.6% or more) [7,14,17,20]. We assume that not only the methodological differences in the assessment of water retention and differences in the quality of the hydrogels themselves, but, first of all, the method of application were the determining factors here. In our studies [3,13], the introduction of pre–swollen, not dry, gels was used, which ensures their uniform distribution and a significant effect from small doses. This most likely explains the stronger effect of the lowering saturated hydraulic conductivity under the influence of small doses of hydrogels up to 20–100 times or more (Figure 9), whereas most sources report only a 2–10 fold decrease, with an exponential dependence on the hydrogel dose [7,10,11,12,13,14]. The possibility of reducing evaporation up to 2–4 times (Figure 11) and a similar prolongation of water consumption of plants without additional watering up to 8–20 days under the influence of superabsorbents was shown earlier in publications [7,10,11,16,18,19]. Information regarding other properties (unsaturated hydraulic conductivity, specific surface area) of gel–soil compositions could not be found; we assume that these properties were only evaluated in our studies [3] due to the new methodological approach. The results obtained for the specific surface area, total water retention energy, and generalized Hamaker constants of sands under the influence of gel–forming soil conditioners were in good agreement with the data for sandy loams and loams, according to [34,35]. In general, our conclusion about the transformation of the hydrophysical properties and fertility of sands to the level of loamy arable soils under the influence of small doses of superabsorbents has been confirmed by other researchers [12,17,19].

#### 4.2.2. Aggregation Strength and Biodegradation of Gel Structures in Soils

Sand particles, due to their low dispersion and surface energy are not capable of self-aggregation. Therefore, they are susceptible to wind erosion (deflation) [6]. The use of 0.1–0.3% gel-forming soil conditioners made it possible to successfully aggregate sandy substrates and achieve an aggregate strength of 40–100 kPa, close to loamy arable soils, including chernozems [36,71]. Obviously, such doses are not suitable for large-scale anti-deflationary fixation of sands, since in terms of 1 hectare, the formation of a surface soil crust of 0.5–1 cm with a bulk density of desert sand of 1.5 g/cm^3^ would require 150–300 kg of dry polymer or 15–30 tons of swollen hydrogel. Therefore, uncrosslinked or weakly crosslinked polyelectrolytes and polycomplexes with a significantly lower cost are more promising for erosion protection [21]. However, gel-forming soil conditioners organically combine improvements in water retention, dispersity, hydraulic properties, anti-pathogenic, and anti-erosion protection of the soil and have no analogues in their effectiveness of complex positive action in soils at such small active doses.

The respiratory assessment of the biological activity of gel-soil compositions and their biodegradability has recently been applied [30], despite the long-term success of this approach in soil science [42,43]. The values of basal respiration, not exceeding 1–2 mg CO_2_/(kg∙h) (Figure 13), indicate the weak biological activity of the gel–soil compositions, according to the criteria existing in soil science [30,43]. However, taking into account the extremely low carbon concentration of hydrogels in the mineral soil substrate (less than 0.1%), this seemingly low respiration for soils corresponds to a high rate of biodegradation (the half-life of superabsorbents was less than 1 year). A comparison with pure hydrogels (see Section 3.1.3.) showed that their mixing with the mineral mass does not reduce, but rather increases, biodegradation, probably due to facilitated microbial colonization of the fragmented gel mass [30]. In any case, acrylic gel-forming soil conditioners are one of the most biodegradable polymeric materials in the soil, which significantly reduces their effectiveness. Usually the organic components of the soil (detritus, humus) have a much higher resistance to biodegradation (with a half-life of tens and hundreds of years) [6,30]. We found a strong effect of the temperature and soil water content on the respiration, and thus the biodegradability of the gel structures (Figure 13). However, in real conditions, especially for arid irrigated agriculture, these factors will not be able to strongly restrain the biodegradation of superabsorbents, since they will always be saturated with water and be at high (25–30 °C and above) temperatures. The incorporation of biocides into the polymer matrix is one of the few factors that actually hinders the biodegradation of composite polymer hydrogels, and our results confirmed their high efficiency with the example of ionic silver.

## 5. Conclusions

This research confirmed the possibility of combining various technological properties in one polymeric gel-forming material, capable of effectively improving water retention, hydraulic conductivity, dispersity and aggregation of soil, its anti-deflationary and anti-pathogenic resistance, along with resistance of gel structures to limiting pedogenic factors (biodegradation, gel collapse due to salinity and temperature, its compression in the pores, etc.). This was facilitated by the optimal ratio of acrylic copolymers, the filling of the polymer matrix by amphiphilic components (dispersed peat, humates), and trace elements and biocides in the form of silver ions and nanoparticles or suitable organic fungicides (for example, Azoxystrobin). The new gel-forming soil conditioners with combined action demonstrated technological properties that have no analogues among known hydrogels, including: an optimal degree of swelling (500–600 kg/kg in pure water and 200–300 kg/kg in saline solutions), high values of hydrophilic specific surface area (600–900 m^2^/g), total surface energy (130–170 kJ/kg), coagulation threshold of gel collapse (80–100 mmol/L), resistance to biodegradation (half-life for at least 5 years under optimal temperature and humidity conditions), suppressing the growth of dangerous plant and soil pathogens (for example late blight), with small effective biocides doses *EC*_50_ of 10–60 ppm. These properties determine a high efficiency from small doses (0.1–0.3%) of these new gel-forming materials, which are capable of increasing the water retention and dispersity of sandy substrates to the level of loams (*FC* = 15–30%, *AWR* = 10–20%, *S*_st_ up to 40–60 m^2^/g), reduce saturated hydraulic conductivity by up to 20–140 times, dually regulate unsaturated hydraulic conductivity, increasing the mobility and availability of water in dry soil and slowing down the capillary-gravitational outflow at high water content, suppress the evaporation rate of water by 2–4 times, form strong soil aggregates (breaking pressure 40–100 kPa), and defend the soil from wind erosion, along with protection of the rhizosphere against pathogens. The results obtained were based on an original system of instrumental methods, models, criteria and gradations, which are convenient for the comparative assessment of polymeric soil conditioners, as presented in this research as a separate methodological part. Despite these significant achievements, the main factors limiting the practical application of gel-forming soil conditioners are the rather high cost and the unsatisfactory resistance of gel structures to osmotic stress and biodegradation. Future efforts in the chemistry of polymer hydrogels for sustainable agriculture and landscaping, in our opinion, should be aimed at solving precisely these problems. Obviously, any new materials in this area must necessarily pass not only laboratory, but also field tests, and we hope in the next publication to present the results of multivariate field trials of new gel-forming soil conditioners in different soil and climatic conditions.

## 6. Patents

The results of the work are used in the synthesis technology of biodegradation-resistant filled hydrogels patented in the Russian Federation:patent RU №2726561 (https://findpatent.ru/patent/272/2726561.html; accessed on 2 July 2022)patent RU 2639789(http://www.findpatent.ru/patent/263/2639789.html; accessed on 2 July 2022).

## Figures and Tables

**Figure 1 polymers-14-04665-f001:**
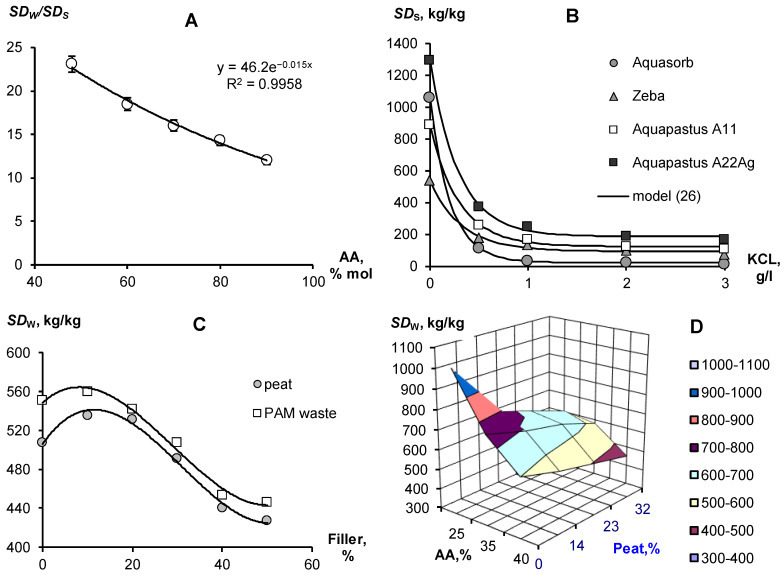
Composition and swelling of polymeric gel-forming materials (explanations in the text). (**A**)— excess of swelling in water relative to KCl solution (3 g/L) depending on the proportion of acrylamide (AA); (**B**)— dependence of the degree of swelling on the concentration of potassium chloride; (**C**)— dependence of the degree of swelling on the content of fillers; (**D**)— effect of the concentration of acrylamide and peat filler on the degree of swelling.

**Figure 2 polymers-14-04665-f002:**
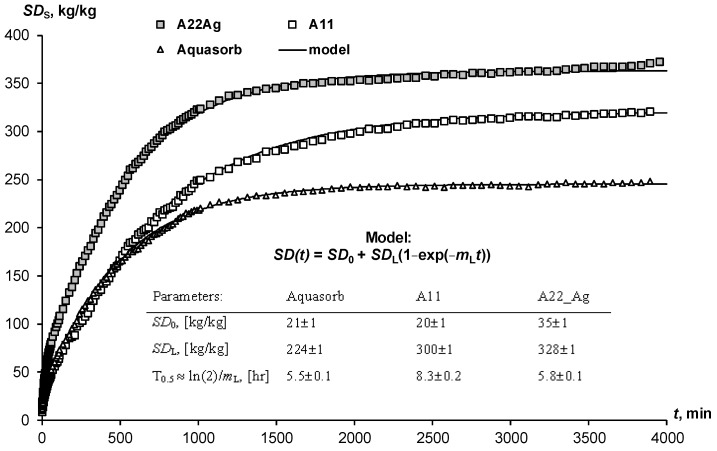
Kinetic curves of limited swelling of hydrogels and their approximation by the Model (27).

**Figure 3 polymers-14-04665-f003:**
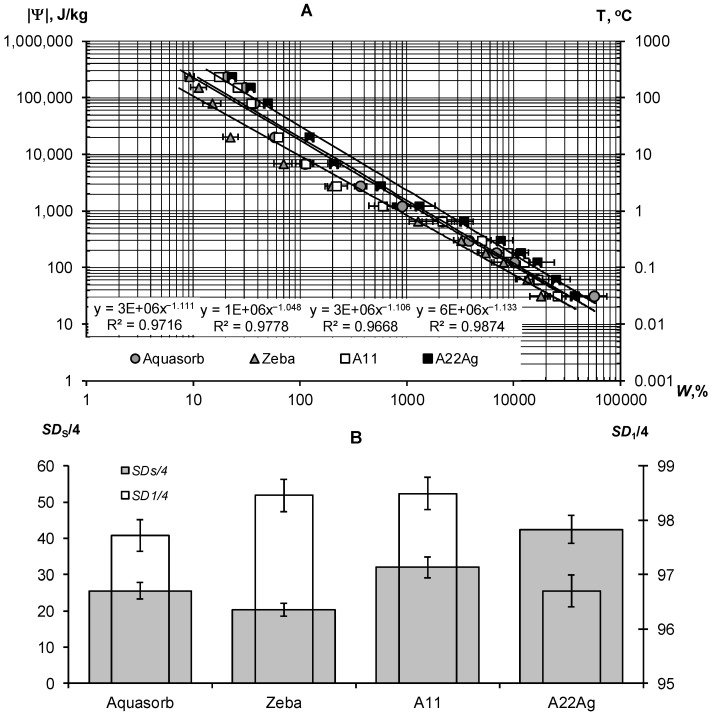
Thermodynamic assessment of water retention in strongly swelled hydrogels: (**A**)—dependence |Ψ|(*W*) and its description using the Campbell–Brooks–Kuri model; (**B**)—calculated quality indicators for compared gels.

**Figure 4 polymers-14-04665-f004:**
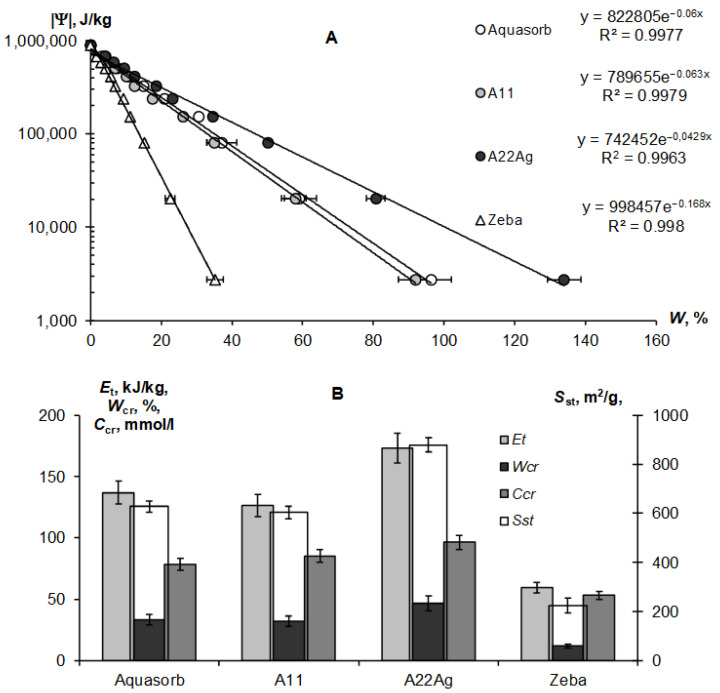
Sorption part of the WRCs for compared hydrogels: (**A**)–experimental data and their approximation using the fundamental Model (6); (**B**)—calculated quality indicators.

**Figure 5 polymers-14-04665-f005:**
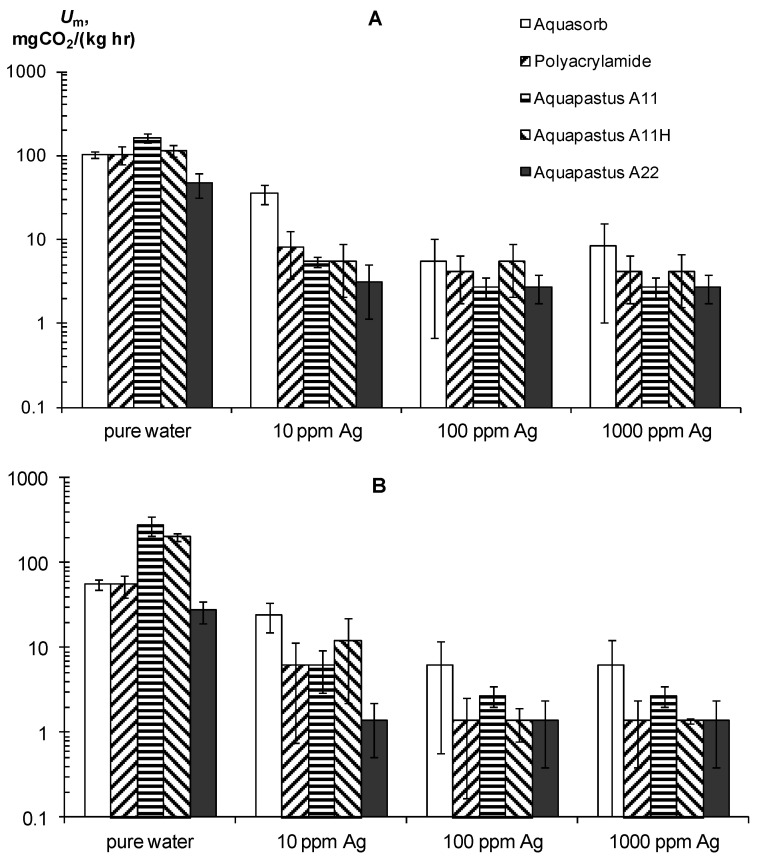
Basal respiration as an indicator of resistance to biodegradation of the hydrogels and their compositions with silver. (**A**)—ionic silver, (**B**)—silver nanoparticles (according to [30]).

**Figure 6 polymers-14-04665-f006:**
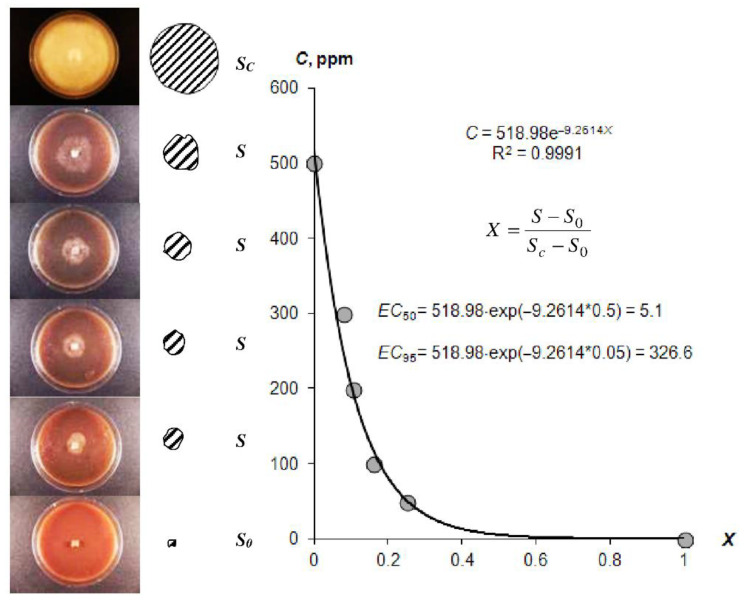
Quantitative approach to the *EC*_50_ and *EC*_95_ assessment according [3] (explanations in the text).

**Figure 7 polymers-14-04665-f007:**
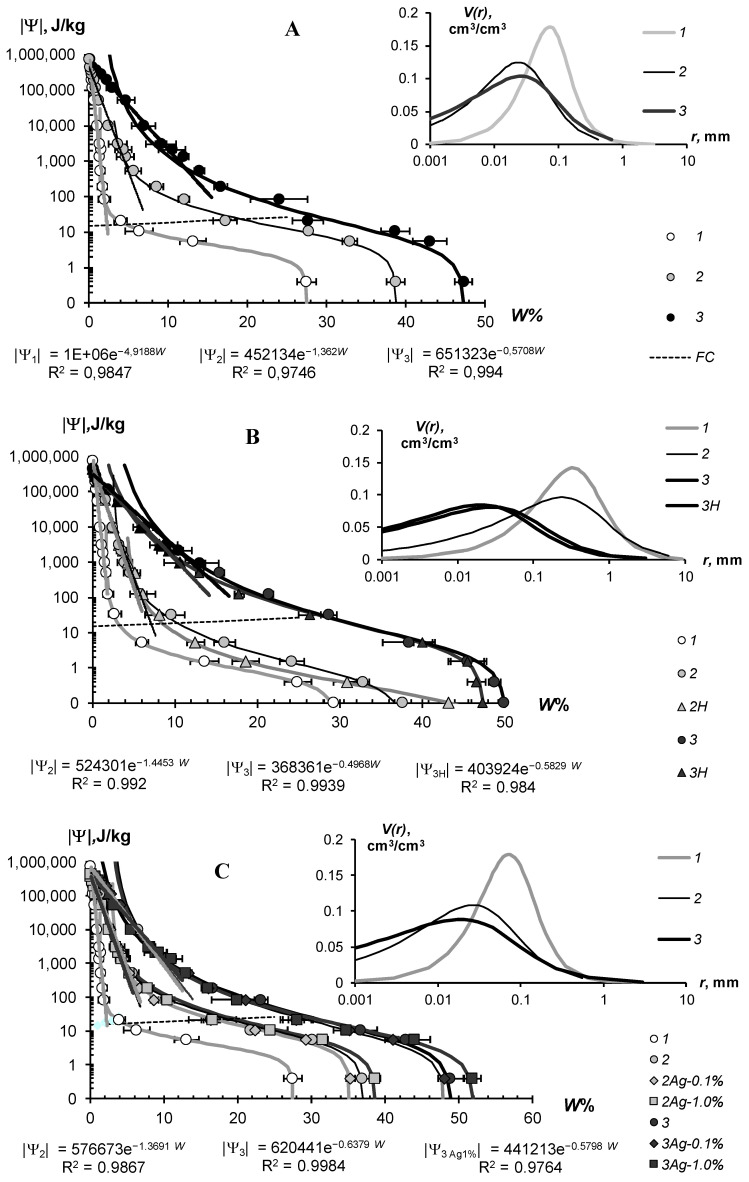
WRC of soil-gel composition (main figure) and pore distribution curves (inset) in monomineral quartz sand. (**A**)—Aquasorb; (**B**)—A11; (**C**)—A22; *1*—untreated control; *2*—0.1% gel dose; *3*—0.3% gel dose; *H*—humates; Ag 0.1%, Ag 1% are silver doses of 0.1 and 1%.

**Figure 8 polymers-14-04665-f008:**
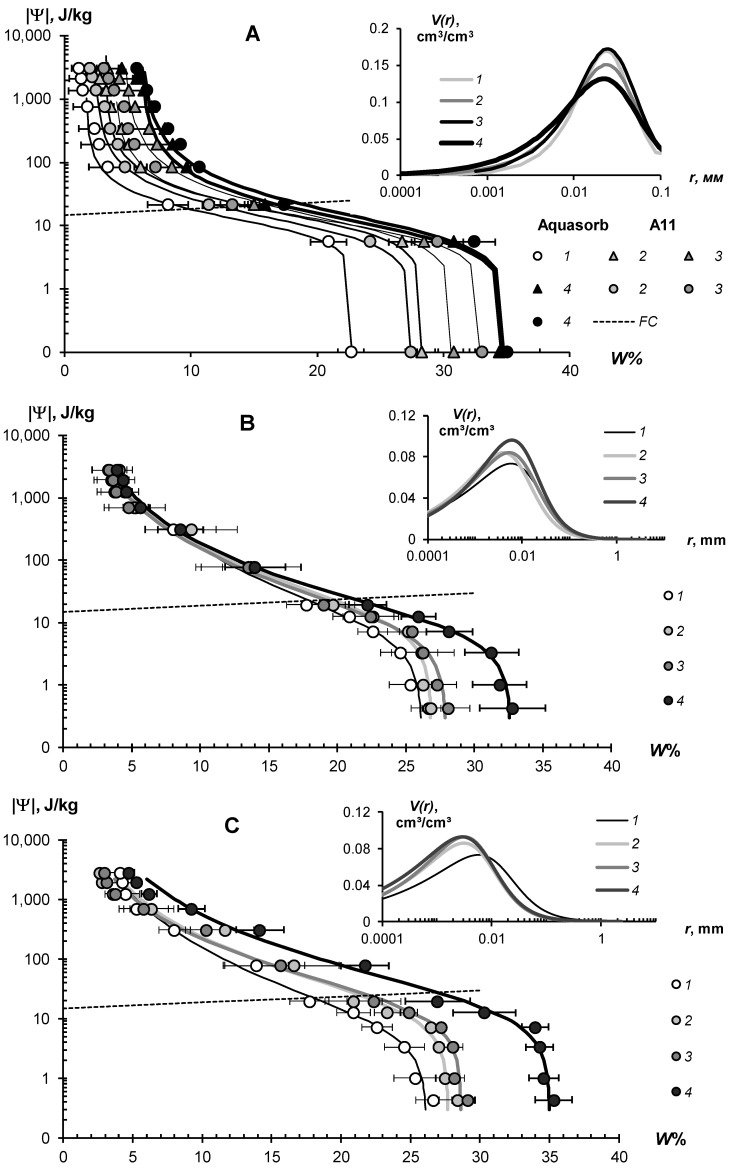
WRC of soil-gel composition (main figure) and pore distribution curves (inset) in loamy-sandy Arenosols. (**A**)—Aquasorb and A11 in Karakum Desert Arenosol; (**B**)—Aquasorb and (**C**)—A22 in Dubai Arenosol; *1*—untreated control; *2*—0.1% gel dose; *3*—0.2% gel dose; *4*—0.2% gel dose.

**Figure 9 polymers-14-04665-f009:**
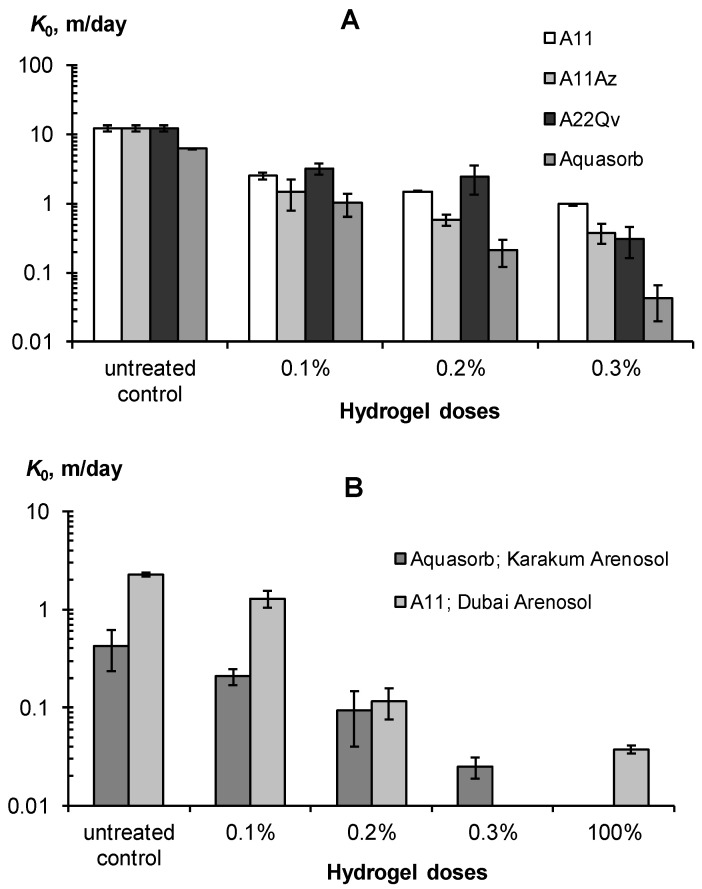
Effect of hydrogels on saturated hydraulic conductivity of coarse-textured soil substrates. (**A**)—monomineral sandy substrates; (**B**)—polymineral loamy-sandy Arenosols.

**Figure 10 polymers-14-04665-f010:**
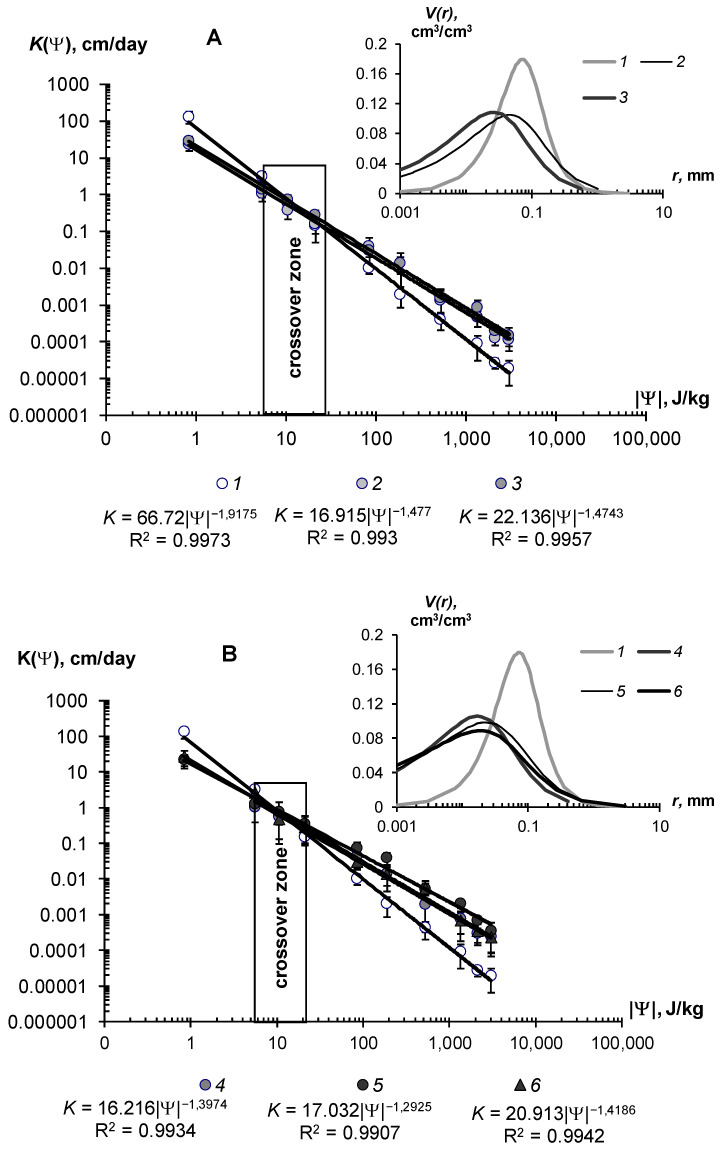
Hydraulic conductivity functions (main figure) and pore distributions (inset) of a monomineral sandy substrate under the influence of hydrogels A22 and A22Ag. (**A**)—small doses; (**B**)—high doses; *1*—mineral substrate (untreated control); doses of hydrogels: *2*—0.05% A22, *3*—0.1% A22, *4*—0.2% A22, *5*—0.3% A22, *6*—0.3% A22Ag. The lines are an approximation of the data by the Campbell [40] model.

**Figure 11 polymers-14-04665-f011:**
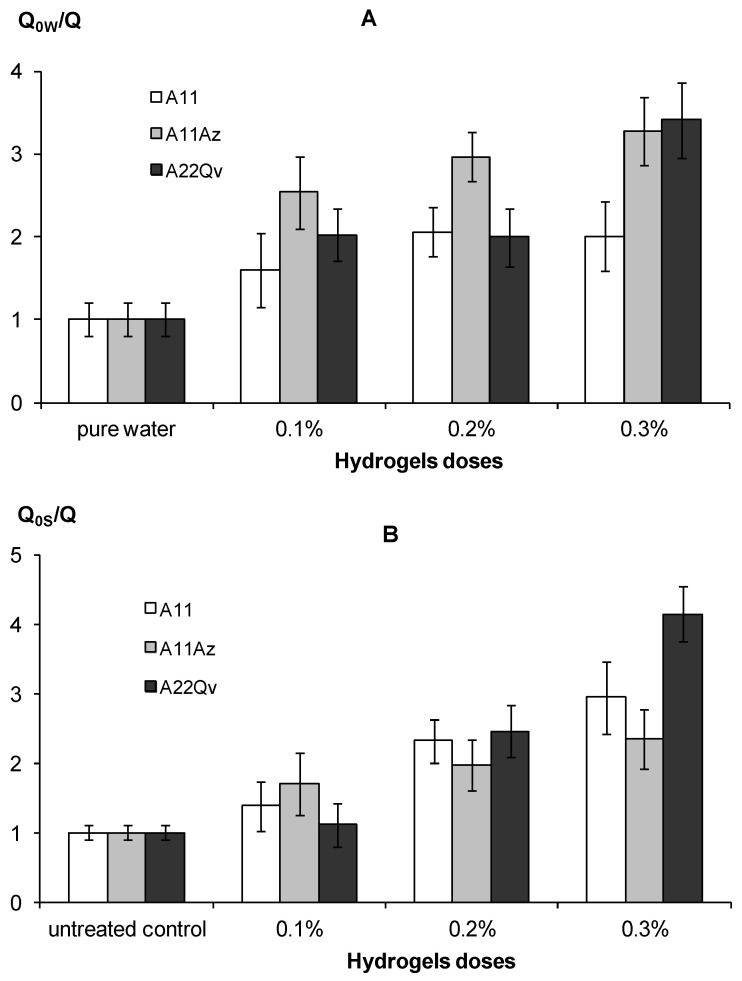
Indicators of a decrease in water evaporation under the influence of gel structures in comparison with pure water (**A**) and sandy substrate (**B**).

**Figure 12 polymers-14-04665-f012:**
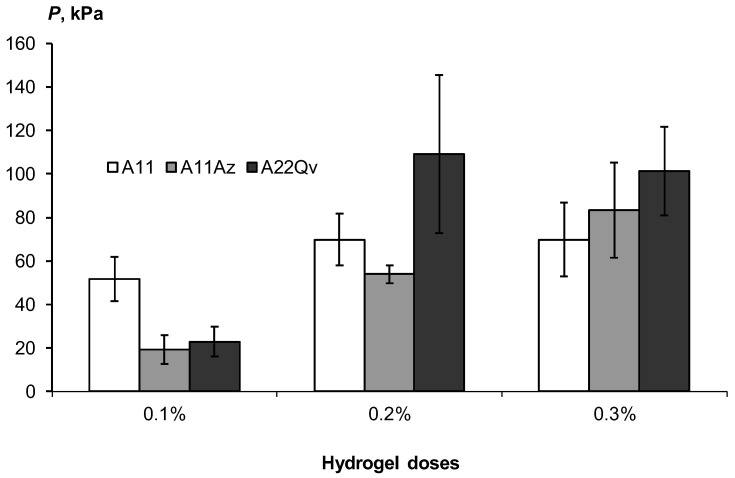
Strength of soil aggregates depending on the dose of gel-forming conditioners.

**Figure 13 polymers-14-04665-f013:**
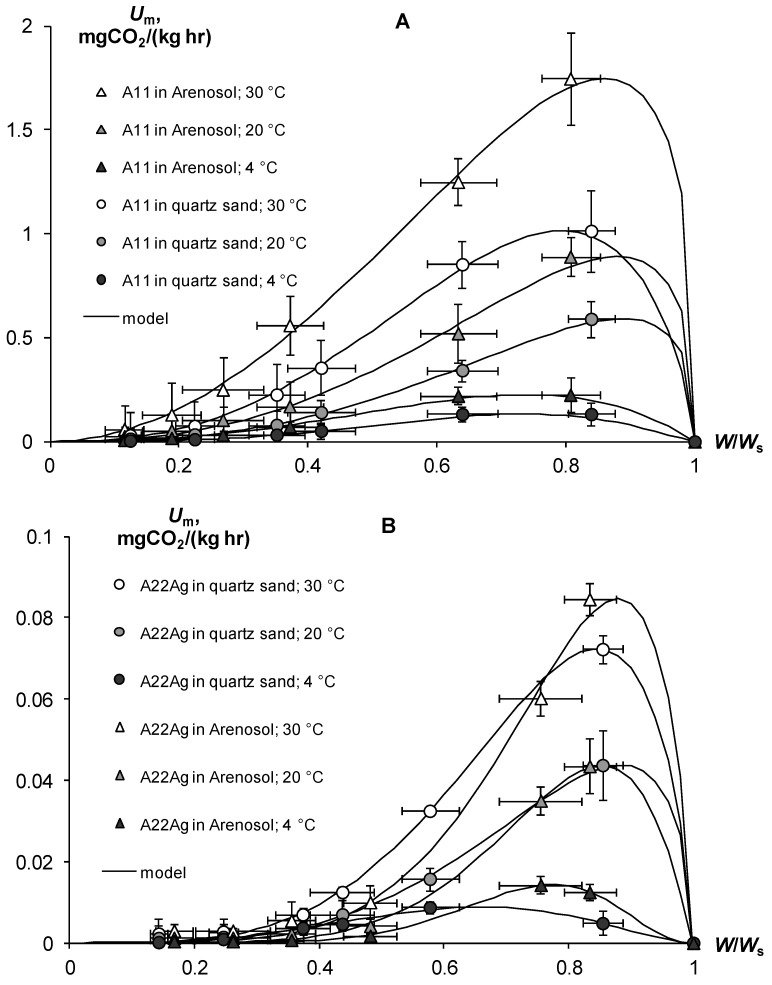
Quantification and modeling of basal respiration of soil-gel compositions (*U*_m_) as a function of temperature and relative water content (*W*/*W_s_*). (**A**)—A11; (**B**)—A22Ag. Model (28) parameters: A11 in Arenosol: 30 °C, *f_(W)_* = (*W*/*W_s_*/0.86)^2.05^{(1 − *W*/*W_s_*)/(1 − 0.86)}^0.33^; R^2^ = 0.998, s = 0.02; 20 °C, *f_(W)_* = (*W*/*W_s_*/0.88)^2.59^{(1 − *W*/*W_s_*)/(1 − 0.88)}^0.35^; R^2^ = 0.992, s = 0.04; 4 °C, *f_(W)_* = (*W*/*W_s_*/0.74)^2.84^{(1 − *W*/*W_s_*)/(1 − 0.74)}^1.00^; R^2^ = 0.992, s = 0.04; A11 in quartz sand: 30 °C, *f_(W)_* = (*W*/*W_s_*/0.79)^2.77^{(1 − *W*/*W_s_*)/(1 − 0.79)}^0.72^; R^2^ = 0.997, s = 0.02; 20 °C, *f_(W)_* = (*W*/*W_s_*/0.89)^2.77^{(1 − *W*/*W_s_*)/(1 − 0.89)}^0.35^; R^2^ = 0.997, s = 0.02; 4 °C, *f_(W)_* = (*W*/*W_s_*/0.73)^3.52^{(1 − *W*/*W_s_*)/(1 − 0.73)}^1.27^; R^2^ = 0.988, s = 0.05; A22Ag in Arenosol: 30 °C, *f_(W)_* = (*W*/*W_s_*/0.88)^5.30^{(1 − *W*/*W_s_*)/(1 − 0.88)}^0.72^; R^2^ = 0.993, s = 0.04; 20 °C, *f_(W)_* = (*W*/*W_s_*/0.85)^6.27^{(1 − *W*/*W_s_*)/(1 − 0.85)}^1.08^; R^2^ = 0.997, s = 0.02; 4 °C, *f_(W)_* = (*W*/*W_s_*/0.77)^8.69^{(1 − *W*/*W_s_*)/(1 − 0.77)}^2.54^; R^2^ = 0.998, s = 0.02; A22Ag in quartz sand: 30 °C, *f_(W)_* = (*W*/*W_s_*/0.75)^4.04^{(1 − *W*/*W_s_*)/(1 − 0.75)}^0.73^; R^2^ = 0.999, s = 0.01; 20 °C, *f_(W)_* = (*W*/*W_s_*/0.88)^3.86^{(1 − *W*/*W_s_*)/(1 − 0.88)}^0.51^; R^2^ = 0.998, s = 0.02; 4 °C, *f_(W)_* = (*W*/*W_s_*/0.66)^3.54^{(1 − *W*/*W_s_*)/(1 − 0.66)}^1.81^; R^2^ = 0.988, s = 0.04.

**Table 1 polymers-14-04665-t001:** Granulometric composition of mineral soil substrates.

Granulometric Fractions:	Soil Substrates:		
	№ 1 – Monomineral Fine-Grained Quartz Sand	№ 2 – Polymineral Loamy-Sandy Arenosol from the Karakum Desert	№ 3– Carbonate Loamy-Sandy Arenosol from the Emirate of Dubai
Clay (<2 µm)	0	2.1	4.2
Silt (2–50 µm)	2.3	23.6	23.3
Very fine sand (50–100 µm)	5.3	23.2	22.4
Fine sand (100–250 µm)	63.8	47.5	39.2
Medium sand (250–500 µm)	28.6	3.6	9.6
Coarse sand (500–1000 µm)	0	0	1.3
Very coarse sand (1000–2000 µm)	0	0	0

**Table 2 polymers-14-04665-t002:** Guidelines for the main quality indicators of gel-forming soil conditioners.

Indicators	Gradations of Quality; Comments		
	«Lack»	«Norm»	«Excess»
*SD*_W_ [kg/kg]	<300 0.1% gel dose does not bind all the water in the soil pores.	300–600 0.1% dose of gel binds all the water in the pores of the soil.	>600 risk of viscous gel flow and its leaching from the soil.
*SD*_S_/4 [%]	<20 poor resistance to pressure in the soil.	20–50 normal resistance to pressure in the soil.	>50 excessive stability; risk of poor water recovery.
Δ*SD*_1_/4 [%]	>98 loosely stitched polymer mesh, risk of gel leaching.	90–98 normal strength of polymer mesh for soil conditioners.	<90 excessively strong polymer mesh, risk of loss of swelling.
*S*_st_ [m^2^/g]	<300 low dispersity, risks of poor water, agrochemicals and pesticides retention in the soil.	300–800 normal dispersity, effective water, agrochemicals and pesticides retention in the soil.	>800 high dispersity, excessively strong water, agrochemicals and pesticides retention in the soil.
*E*_t_ [J/kg]	<6000 low water retention, sorption capacity and structuring effect in the soil.	60,000–160,000 normal surface energy of hydrogels providing optimal technological properties in the soil.	>160,000 excessively high surface energy; risks of poor aeration and clumpiness in the soil.
*C*_cr_*W*_cr_ [mmol/kg] (if z = 1)	<20 low coagulation threshold; risk of loss of aggregate stability during drying and freezing of gels	20–40 moderate coagulation threshold; normal aggregate stability and recovery during drying and freezing of gels	>40 high coagulation threshold; high aggregative stability and resistance to drying and freezing of gels
*T*_0.5_ [yr]	<2 low biodegradable stability; complex use of gels in soils is not cost-effective	2–10 stability of gels acceptable for their effective use in the soil	>10 increased biodegradable stability, polymers cannot be used in control release systems
*EC*_50_ [ppm]	>200 Insufficient suppression of pathogens in the rhizosphere	20–200 normal effectiveness of pathogen suppression in the rhizosphere	<20 excessive dose of biocides, risk of damage to plants and pedofauna of the rhizosphere

**Table 3 polymers-14-04665-t003:** Guidelines for the main quality indicators of gel compositions with coarse-textured soil substrates (doses range of hydrogels 0.1–0.3%).

Indicators	Gradations of Quality; Comments		
	«Lack»	«Norm»	«Excess»
γ_1_ = *S*_st_/*S*_st_^0^	<2 weak increase in dispersity; poor quality or too small dose of soil conditioner.	2–6 normal increase in dispersity; good structuring and water-retaining effect.	>6 strong increase in dispersity; high quality gels, doses can be reduced
γ_2_ = *E*_t_/*E*_t_^0^	<1.5 weak increase in surface energy; poor quality or too small dose of soil conditioner.	1.5–3 normal increase in surface energy; good technological properties of soil conditioners	>3 strong increase in surface energy; high quality gels, doses can be reduced
γ_3_ = (*A*_G_^0^/*A*_G_)^2^	<2 weak aggregative stability, high risk of gel collapse under salinity	2–4 normal aggregative stability; gels can be used in slightly saline soils	>4 high aggregative stability; gels are suitable for moderately saline soils and water
*FC* [%]	<10 low water capacity at the level of sandy substrates; poor quality or too small dose of soil conditioner	10–20 normal water capacity at the level of sandy loams and loams; good quality gels	>20 high water capacity at the level silty and clayey loams; very good quality gels, doses can be reduced
*AWR* = *FC* − *WP* [%]	<8 lack of available water, poor effect of soil conditioners	8–16 normal range for fertile soil, good effect from soil conditioners	>16 wide range of available water, high quality gels, doses can be reduced
*K*_0_ [m/day]	<0.3 low water conductivity, stagnant water, risk of poor soil aeration	0.3–1 normal water conductivity, acceptable infiltration losses, good aeration and gas exchange with the atmosphere	>1 high water conductivity, large infiltration water losses
*K* _0S_ */K* _0_	<5 rather weak decrease in water conductivity; poor quality or too small dose of soil conditioner	5–40 effective reduction of infiltration; good quality air conditioner	>40 excessive reduction in infiltration; risks of stagnant water and low soil aeration
*Q*_0W_*/Q* or *Q*_0S_/*Q* within 5 days	<1.5 insufficient evaporation reduction.	1.5–3 normal evaporation reduction efficiency.	>3 high evaporation reduction efficiency
*P*_A_ [Pa]	<50 low particle consolidation, weak soil resistance to erosion	50–100 normal particle consolidation and soil resistance to erosion	>100 high particle consolidation and soil resistance to erosion

**Table 4 polymers-14-04665-t004:** The effective concentration for two-fold and total suppression of late blight growth.

Hydrogels/Treatment	*EC* _50_	*EC* _95_
Ionic silver		
Aquasorb	42.0 ± * 4.4	448 ± 31
A11	63.0 ± 19.6	467 ± 29
A22	17.0 ± 5.4	406 ± 44
LSD_0.05_ ^*^	24.0	70.6
LSD_0.01_ ^*^	36.4	106.9
Silver nanoparticles		
Aquasorb	1.4 ± 0.2	264 ± 13
A11	0.9 ± 0.1	272 ± 80
A22	5.8 ± 2.4	241 ± 90
LSD_0.05_	2.8	139.9
LSD_0.01_	4.2	211.8
Azoxystrobin in sythetic fungicide Quadris		
Aquasorb	14.1 ± 3.3	237 ± 18
A11	4.9 ± 3.1	222 ± 10
A22	17.3 ± 6.5	187 ± 13
LSD_0.05_	9.1	28.1
LSD_0.01_	13.9	42.6

* Hereinafter, ± means the boundaries of the confidence interval at *p* < 0.05 significance level, LSD_0.05_ and LSD_0.01_ are the Least Significant Differences at *p* = 0.05 and *p* = 0.01 significance levels, respectively.

**Table 5 polymers-14-04665-t005:** Estimated parameters of water retention and dispersity in mineral-gel compositions with quartz sand.

Hydrogels, Doses %	*FC*, %	*WP*, %	*AWR*,%	*W*_s_, %	*W*_r_, %	α, kPa^−1^	*n*	*E*_t_, J/kg	*S*_st_, m^2^/g	*A*_G_, 10^−19^ J
Monomineral quartz sands:										
0%	4.6	1.3	3.3	27.5	1.3	0.310	2.26	2033	7.4	3.96
0%	3.3	1.7	1.6	29.3	1.6	1.655	1.87	1973	7.2	3.95
0%	3.0	0.6	2.4	26.4	0.6	0.523	2.11	727	3.1	3.22
Aquasorb										
0.1%	19.2	4.2	15.0	38.7	2.8	0.146	1.61	3320	26.6	1.79
0.2%	26.0	8.3	17.6	49.2	3.8	0.209	1.40	6928	46.8	2.12
0.3%	28.5	9.8	18.7	48.6	1.4	0.185	1.30	11410	63.5	2.58
A22 (23% Peat)										
0.1%	18.6	4.0	14.5	37.0	1.8	0.183	1.50	4212	26.5	2.28
0.1% + Ag 0.1%	17.0	4.1	12.9	35.0	2.9	0.169	1.61	3573	28.2	1.82
0.1% + Ag 1%	19.5	4.1	15.3	38.6	1.2	0.199	1.45	3050	26.0	1.68
0.2%	23.7	5.8	17.9	40.7	2.0	0.130	1.44	5073	36.1	2.02
0.2% + Ag 0.1%	22.7	6.6	16.1	39.6	1.7	0.188	1.36	5836	42.5	1.97
0.2% + Ag 1%	21.6	6.9	14.6	38.6	5.6	0.118	1.63	7685	44.4	2.49
0.3%	28.6	9.5	19.1	51.9	2.7	0.213	1.34	9726	56.8	2.46
0.3% + Ag 0.1%	27.4	9.9	17.5	48.2	2.8	0.232	1.32	1154	53.6	2.97
0.3% + Ag 1%	29.0	9.6	19.4	48.9	0.1	0.203	1.29	8206	61.1	1.93
A11 (28% biocatalytic waste)										
0.1%	10.2	3.6	6.6	37.6	2.4	1.913	1.42	3753	22.6	2.38
0.2%	20.7	9.6	11.1	53.0	0.2	6.459	1.19	5630	62.7	1.29
0.3%	28.4	11.4	17.0	50.0	1.9	0.323	1.26	7415	72.9	1.46
A11H (12% Humates)										
0.1%	8.6	4.5	4.1	50.0	3.9	7.257	1.47	3822	24.8	2.22
0.2%	16.7	7.2	9.5	48.6	0.1	9.817	1.20	5652	45.7	1.74
0.3%	28.1	9.7	18.3	47.5	0.0	0.230	1.27	6929	62.2	1.60
A11HMZ (12% Humates, Zn, Mg 0.4%)										
0.1%	9.4	4.6	4.8	57.0	3.9	7.337	1.46	4561	25.6	2.56
0.2%	15.2	6.2	9.0	53.0	1.9	7.389	1.27	4587	39.1	1.68
0.3%	29.7	9.5	20.2	53.3	0.2	0.231	1.29	6666	56.6	1.69
A11Ag (ionic silver 1%)										
0.1%	10.9	2.8	8.2	29.8	0.0	1.234	1.32	4025	21.5	2.69
0.2%	17.2	5.6	11.6	42.6	0.0	1.360	1.27	7251	38.2	2.72
0.3%	27.0	9.7	17.3	57.8	0.0	0.688	1.26	12547	68.4	3.26
A11Az (10% Filterperlite, Azoxystorobin 1%)										
0.1%	9.5	1.8	7.7	33.4	0.3	1.206	1.41	3303	16.6	2.86
0.2%	16.1	4.7	11.4	41.8	0.0	1.203	1.29	6987	34.2	2.94
0.3%	23.8	8.6	15.3	51.6	0.0	0.823	1.25	12553	56.8	3.17
A22Qv (23% Peat, Quadris 1%)										
0.1%	10.3	2.1	8.2	34.6	0.3	1.227	1.39	3951	17.6	3.23
0.2%	19.1	7.1	11.9	48.1	0.0	2.195	1.24	10326	44.4	3.32
0.3%	25.4	9.4	16.0	52.6	0.0	0.705	1.25	14737	64.2	3.29

Hereinafter, ***W*_s_**, ***W*_r_**, α, ***n*** are the van-Genuchten model parameters (see Formula (4)).

**Table 6 polymers-14-04665-t006:** Estimated parameters of water retention in mineral-gel compositions with loamy sandy Arenosols.

Hydrogels, doses %	*FC*, %	*WP*, %	*AWR*,%	*W*_s_, %	*W*_r_, %	α, kPa^−1^	*n*	R^2^	s, %	*p*-Value
Loamy-sandy Arenosol from the Karakum Desert, Aquasorb										
0%	9.8	1.7	8.0	22.9	1.7	0.093	2.11	0.997	0.6	0.0005
0.1%	12.5	2.6	9.9	27.4	2.5	0.111	2.06	0.998	0.6	0.0001
0.2%	14.4	4.1	10.3	33.0	4.0	0.110	2.16	0.996	0.8	0.0002
0.3%	18.4	6.5	12.0	35.1	6.1	0.112	1.83	0.995	1.0	0.0015
Loamy-sandy Arenosol from the Karakum Desert, A11										
0%	9.8	1.7	8.0	22.9	1.7	0.093	2.11	0.997	0.6	0.0005
0.1%	14.3	3.3	11.0	28.3	3.2	0.089	2.14	0.996	0.8	0.0007
0.2%	16.0	5.0	11.0	30.8	4.8	0.105	1.92	0.993	1.0	0.0030
0.3%	16.8	6.3	10.5	34.8	6.0	0.123	1.93	0.993	1.1	0.0023
Loamy-sandy Arenosol from Dubai, Aquasorb										
0%	18.0	4.7	13.3	26.1	0.0	0.111	1.34	0.995	0.7	0.0050 *
0.1%	19.7	4.2	15.5	26.8	0.0	0.069	1.40	0.992	1.6	0.0240 *
0.2%	19.2	4.1	15.1	27.9	0.0	0.091	1.39	0.995	0.9	0.0008 *
0.3%	21.2	4.6	16.6	32.6	0.9	0.097	1.43	0.992	1.4	0.0051 *
Loamy-sandy Arenosol from Dubai, A22										
0%	18.0	4.7	13.3	26.1	0.0	0.111	1.34	0.995	0.7	0.0050 *
0.1%	21.5	4.8	16.7	27.6	0.0	0.053	1.41	0.997	0.8	0.0005 *
0.2%	22.1	4.3	17.7	28.6	0.0	0.051	1.44	0.996	0.8	0.0019 *
0.3%	26.7	6.9	20.0	35.0	0.0	0.052	1.37	0.999	0.4	0.0001 *

* The highest *p*-value, excluding the *W*r parameter, for which the estimate was often not statistically reliable.

**Table 7 polymers-14-04665-t007:** Biodegradation constants and half-lives of hydrogels at optimal humidity for soil-gel compositions.

Indicators:	*C*_org_, %	Temperature:		
		4 °C	20 °C	30 °C
A11 in quartz sand				
*k*_0_, yr^−^^1^	0.071	0.24	1.09	1.88
*T*_0.5_, yr		2.8	0.6	0.4
A22Ag in quartz sand				
*k*_0_, yr^−^^1^	0.070	0.01	0.08	0.14
*T*_0.5_, yr		76.4	8.5	5.1
A11in loamy-sandy Arenosol (UAE, Dubai)				
*k*_0_, yr^−^^1^	0.098	0.30	1.19	2.35
*T*_0.5_, yr		2.3	0.6	0.3
A22Ag in loamy-sandy Arenosol (UAE, Dubai)				
*k*_0_, yr^−^^1^	0.088	0.02	0.06	0.13
*T*_0.5_, yr		37.3	10.7	5.5

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
