# Peer review of "Gel-Forming Soil Conditioners of Combined Action: Laboratory Tests for Functionality and Stability"

_polymers, 2022, doi:10.3390/polym14214665_

Round 1

Reviewer 1 Report

This work is a complete scientific study in applied soil physics, interesting from the point of view of the applied technology for the conservation of soil properties due to gel injection. The method proposed by the authors also raises a number of fundamental questions for soil physics, which were solved by the authors of the article in the course of the study. 

The paper analyzes the treatment of soils with gel solutions in order to improve its water-retaining and rheological properties. The former contribute to an increase and longer retention of moisture in soils, the latter affect the reduction of soil erosion. The gel, which is pumped into the soil, is highly resistant to natural and man-made soil-destroying factors. Osmotic swelling, compaction (consolidation) of soils, and biodegradation of the gel are considered among such factors. The complex of laboratory research methods is based on the system of tests developed by the authors. The paper presents mathematical models that describe the process of functioning of soils treated with gel, on the basis of which criteria for the quality of gels and their compositions with mineral soil substrates are developed. New materials have technologically optimal degree of swelling (200-600 kg/kg in pure water and saline solutions with TDS 1-3 g/l), high values ​​of surface energy (>130 kJ/kg), specific surface area (>600 m2/g ), gel breakdown threshold (>80 mmol/l), half-life (>5 years), powerful antifungal and antibiotic activity (doses of EC50 biocides 10-60 ppm). Due to these properties, new gel-forming materials in small doses of 0.1-0.3% increase the water-retaining capacity and dispersion of sandy substrates to the level of loams, reduce saturated hydraulic conductivity by 20-140 times, and suppress evaporation by 2-4 times. times, form a windproof soil crust (strength up to 100 kPa). The article is of scientific and practical interest and may be published in the journal Polymers.

Author Response

Dear and highly respected Reviewer #1! Thank you very much for your positive assessment of the manuscript and permission to publish it in Polymers. We have corrected the text, eliminating typos and grammatical errors in the revised version of the manuscript.

From the team of authors with best wishes, Prof. Andrey V. Smagin

Reviewer 2 Report

Dear authors

The main question addressed by the research is the analysis of the technological properties and stability of innovative gel-forming polymeric materials for complex soil conditioning. This is very relevant and interesting for the readers taking into account that these kinds of information and analysis used in the evaluation process for the applicability of complex soil conditioners have not been in one protocol before. This point presents the originality of the topic. The main addition of the manuscript to the subject area compared with other published materials is the following items;

a)        The developed soil conditioner materials combine the improvement of water retention, dispersity, hydraulic properties, anti-erosion and anti-pathogenic protection of the soil along with high resistance to negative environmental factors (osmotic stress, compression in the pores, microbial biodegradation).

b)       Presents Laboratory analysis based on an original system of instrumental methods, new mathematical models, criteria, and gradations of the quality of gels and their compositions with mineral soil substrates.

c)        The developed soil conditioner materials have a technologically optimal degree of 19 swellings (200-600 kg/kg in pure water and saline solutions with 1-3 g/l TDS), high values of surface 20 energy (>130 kJ/kg), specific surface area (>600 m2/g), the threshold of gel collapse (>80 mmol/l), half-21 life (> 5 years), powerful fungicidal effect (EС50 biocides doses of 10-60 ppm).

d)       The new gel-forming materials in small doses of 0.1-0.3% increase the water retention and dispersity of sandy substrates to the level of loams, reduce saturated hydraulic conductivity by 20-140 times, suppress the evaporation by 2-4 times, form windproof soil crust (strength up to 100 kPa).

e)       New methodological developments and recommendations are useful for complex laboratory testing of hydrogels in small (5-10 g) soil samples.

f)        Presents a solution to the main drawbacks of the developed commercial soil conditioner hydrogels especially the reduced water holding capacity in saline water.

The paper is well written, however, in some places of the text repeated discussion and theoretical explanations have been found in comparison with other published results. This makes the paper somehow lengthy, hard to follow, and looks like a review more than a research article. Shortening the manuscript in this direction is recommended to facilitate the reading of the paper. On the other hand, the delivered conclusions are consistent with the evidence and presented arguments.

In conclusion, the authors addressed satisfactorily the posed question and enrich the field with needed information in this regard. 

Greetings

Author Response

Dear and highly respected Reviewer #2! Thank you very much for your positive assessment of the manuscript at the stage of its preparation in Polymers. We have corrected the text, eliminating typos, repetitions and grammatical errors in the revised version of the manuscript. However, unfortunately, we cannot fulfill your proposal to shorten the text, since the Reviewers of the first, shorter version of the manuscript (July 7 submission), on the contrary, obliged us to supplement the Introduction and divide the Results and Discussion into two separate sections. This led to an increase in the volume of the article and the need for a consistent return in the "Discussion" section to the Tables and Figures from the "Results", during comparison with known published data. We hope that this explanation will be satisfactory for you, since Polimers has adopted a mandatory separation of Results and Discussion.

From the team of authors with gratitude and best wishes,

Prof. Andrey V. Smagin

Reviewer 3 Report

Dear Authors,

I reviewed your article titled (Gel-forming Soil Conditioners of Combined Action: Labora-2 tory Tests for Functionality and Stability). Overall, the data presented here is valuable to those working in this field and demonstrates the effectiveness of a relatively simple intervention that could be applied a wider scale, especially in the field of soil science. The article is written very well and organized. The introduction part is written well and related to the subject of this article. The materials and methods are clear and enough and support the hypothesis. The results are the most interesting part that has precise data. The discussion part also is descriptive. Thus, I suggest publishing this article in its current form. 

All the best

Author Response

Dear and highly respected Reviewer #3! Thank you very much for your positive assessment of the manuscript and the decision on the possibility of its publication in Polymers in the current form.

From the author's team, sincerely

Prof. Andrey V. Smagin